# Tree-Structured Orthonormal Decomposition of the Aitchison Simplex

**Daisuke Yamada** [1]  **Qijun Zhang** [1]  **Travis Pence** [1]  **Barbara B. Bendlin** [1]  **Federico Rey** [1]  **Vikas Singh** [1]

## Abstract

Compositional data—vectors encoding relative proportions—arise across scientific domains, including ecology, geochemistry, and genomics. The features in these data often come with known hierarchical structure (e.g., taxonomies, phylogenies, ontologies), yet existing methods either ignore this structure, discard the intrinsic Aitchison geometry, are designed for binary trees, or yield incomplete coordinate systems. We describe *PolyILR*, a canonical orthonormal decomposition of the Aitchison tangent space aligned with any tree topology. Our construction defines a weighted local geometry at each internal node capturing full branching structure, then lifts these to a global orthonormal basis where every coordinate corresponds to a specific tree location. On microbiome and single-cell benchmarks, PolyILR yields stable, interpretable features and enables inference at multiscale tree resolution. We also establish a novel theoretical connection to softmax classifiers, suggesting possible applications to probabilistic modeling.

## 1. Introduction

*Compositional data*—nonnegative vectors (or components) whose summed total carries no intrinsic meaning—arise across scientific and statistical settings, including microbiome profiles, cell type proportions, ecological counts, and probabilistic model outputs (Gloor et al., 2017; Billheimer et al., 2001; Buettner et al., 2021). Such data live on the simplex, where inference is based on *relative* not absolute values. Often, the components are not exchangeable and are organized by domain hierarchies reflecting evolutionary, functional, or semantic relationships (Silverman et al., 2017; Harmon, 2019). These trees describe which comparisons among components are meaningful and at what resolution.

The standard tool in compositional data analysis (CoDA) is *Aitchison geometry* (Aitchison, 1982), which formalizes that only ratios are informative. This is achieved via an isometric log-ratio (ILR) embedding of compositional data into Euclidean space (Egozcue et al., 2003). Here, one treats components *symmetrically*, partly reflecting origins in domains where no external structure was assumed (Egozcue & Pawlowsky-Glahn, 2005; Mandal et al., 2015). But in many applications, domain-specific hierarchies are known: trees specify which comparisons are meaningful and how they are related. To address this gap, the literature provides strategies to incorporate tree structure, such as tree-based balances and phylogeny-aware coordinates. Doing so improves interpretability and downstream analysis. Yet, existing approaches typically address specific regimes (e.g., binary trees as in phylogenetic reconstruction, selected contrasts) (Silverman et al., 2017; Washburne et al., 2017; Morton et al., 2017) or rely on construction choices *external* to the geometry (Lozupone & Knight, 2005; Mao & Ma, 2022). For instance, a polytomous node admits no canonical binary resolution, yet binary-tree methods force an arbitrary choice (Figure 1). Hence, the key question we study is whether hierarchies can be incorporated in a way that is general, geometrically principled, and canonical for arbitrary trees.

**Main difficulty.** The core issue is structural incompatibility. Aitchison geometry identifies compositions up to a *global* scaling: a $(d-1)$-dimensional Euclidean tangent space where ILR coordinates live. But trees impose *local* and *nested* constraints: distinctions are meaningful within clades (i.e., subtrees) and comparisons happen at multiple resolutions. Aligning these requires *decomposing* the Aitchison tangent space to respect branching structure at every internal node. Binary trees sidestep this issue by reducing each node to a single contrast (Figure 1). For multi-branching (i.e., *polytomous*) hierarchies, no canonical construction exists.

**This paper.** We ask if an orthonormal decomposition of the Aitchison tangent space can be compatible with arbitrary tree structure and the simplex invariances—without sacrificing isometry or introducing arbitrary choices. Such a decomposition would give each internal node a geometrically motivated signature and provide a multiscale coordinate system for simplex-valued data on the same footing as standard ILR methods (Pawlowsky-Glahn & Egozcue, 2001), while grounding the analysis firmly in tree structure.

[1]University of Wisconsin Madison, Madison WI, USA. Correspondence to: Daisuke Yamada <dyamada2@wisc.edu>.

*Proceedings of the 43$^{rd}$ International Conference on Machine Learning*, Seoul, South Korea. PMLR 306, 2026. Copyright 2026 by the author(s).

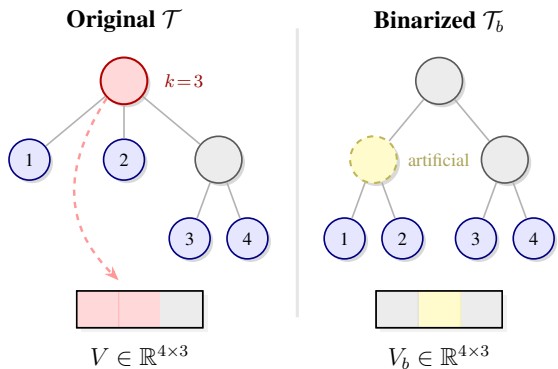

**Original $\mathcal{T}$**      **Binarized $\mathcal{T}_b$**

$V \in \mathbb{R}^{4 \times 3}$      $V_b \in \mathbb{R}^{4 \times 3}$

*Figure 1. Polytomous vs. Binarized Tree.* (Left) $\mathcal{T}$ with a polytomous root ($k = 3$). PolyILR assigns $k - 1 = 2$ basis vectors (column of $V$) to the root (red). (Right) Binarized $\mathcal{T}_b$ required by PhILR (*one coordinate per node*) introduces an artificial node (yellow) encoding an arbitrary grouping of leaves 1 and 2—a choice not justified by original $\mathcal{T}$.

The absence of such a coordinate system extends beyond traditional CoDA to probabilistic model representations. Softmax outputs are simplex-valued but often represented in flat probability coordinates, while structure over outcomes is increasingly explicit: class taxonomies, semantic hierarchies, grammars, tree-structured search spaces (Silla Jr & Freitas, 2011). The absence of a canonical tree-aligned coordinate system beyond heuristics limits analysis of where probability mass, errors, or learning signals concentrate.

**Contributions.** We (1) construct **PolyILR** (Polytomous ILR), a canonical orthonormal decomposition of the Aitchison tangent space aligned with arbitrary trees, answering affirmatively the question raised above, (2) demonstrate its utility in CoDA; stable feature selection and tree-level inference in standard microbiome and single-cell datasets, (3) establish a novel theoretical connection to softmax classifiers via shared invariance structure. Our goal is interpretability of the representation itself, not downstream performance: each coordinate corresponds to a specific tree location (i.e., a node-contrast pair) that yields consistent, tree-grounded analysis and inference (see Table 1). Code is available at https://github.com/vsingh-group/polyilr.

**Conflict of Interest Disclosure.** The authors declare no financial conflicts of interest related to this work.

## 2. Background

We introduce the geometry of compositional data and formalize the tree alignment problem (see Aitchison (1982)).

### 2.1. Aitchison Geometry

**Compositional data.** Compositional data are nonnegative vectors whose totals are uninformative and only relative proportions matter, e.g., microbial abundances (counts with varying sequencing depth) or chemical concentrations (parts of a mixture) (Gloor et al., 2017; Jackson, 1997). We normalize to unit sum, placing data in the open simplex:

$$\Delta^{d-1} = \left\{ x \in \mathbb{R}^d_{>0} : \sum_{i=1}^d x_i = 1 \right\}. \tag{1}$$

The key constraint is that only ratios $x_i/x_j$ carry information, not absolute values.

**Geometry is fixed.** Aitchison geometry (Aitchison, 1982) formalizes this by equipping the set $\Delta^{d-1}$ with perturbation $x \oplus y = \mathcal{C}(x_1 y_1, \ldots, x_d y_d)$ and powering $\alpha \odot x = \mathcal{C}(x_1^\alpha, \ldots, x_d^\alpha)$, where $\mathcal{C}(\cdot)$ is the closure. Under these operations, $(\Delta^{d-1}, \oplus, \odot)$ forms a $(d-1)$-dimensional Hilbert space with inner product:

$$\langle x, y \rangle_A = \frac{1}{d} \sum_{i=1}^d \sum_{j=1}^d \log \frac{x_i}{x_j} \log \frac{y_i}{y_j}.$$

The induced Aitchison distance $d_A(x, y) = \|x \ominus y\|_A$ is perturbation-invariant: $d_A(x \oplus z, y \oplus z) = d_A(x, y)$. Note that this geometry is *not a modeling choice*—it is the unique structure respecting compositional invariance.

### 2.2. ILR Basis

**Basis is a choice.** The centered log-ratio (CLR) transform maps $x \in \Delta^{d-1}$ (under Aitchison geometry) isometrically to the CLR hyperplane (i.e., *Aitchison tangent space*)

$$\mathcal{H} = \{ z \in \mathbb{R}^d : \mathbf{1}^\top z = 0 \} \tag{2}$$

via $\text{clr}(x) = (\log x_1/g(x), \ldots, \log x_d/g(x))$, where $g(x) = (\prod_i x_i)^{1/d}$ is the geometric mean. Any matrix $V \in \mathbb{R}^{d \times d-1}$ whose *columns form an orthonormal basis* of $\mathcal{H}$ yields isometric log-ratio (ILR) coordinates and admits an isometric bijection

$$\varphi(x) = V^\top \log x, \tag{3}$$

from $(\Delta^{d-1}, \langle \cdot, \cdot \rangle_A)$ to $(\mathbb{R}^{d-1}, \langle \cdot, \cdot \rangle_2)$ (Egozcue et al., 2003). All such bases $V$ are related by orthogonal transformations. That is, they induce the same geometry but different decompositions of the simplex. The question is thus not whether to use an ILR basis, but *which basis to choose*, and this choice largely determines interpretability.

**Basis choice controls interpretability.** This parallels classical signal processing: Fourier bases yield frequency components from translation symmetry (Brigham, 1988); wavelet bases yield scale-localized components from dyadic partitions (Mallat, 2002). Aligning the basis with domain structure produces interpretable coordinates.

# 3. Problem Setup

**Compositions come with tree.** The $d$ components of a composition are often organized by a known rooted tree $\mathcal{T}$, e.g., phylogenetic or taxonomic trees in ecology, gene ontologies in genomics (Ashburner et al., 2000; Lozupone & Knight, 2005). The tree encodes domain structure: which comparisons are meaningful and at what resolution. Hence, we seek a basis $V$ aligned with $\mathcal{T}$.

**Binary trees.** When $\mathcal{T}$ is binary, each internal node $u$ has exactly two children, yielding one *contrast*: the log-ratio of geometric means of the two descendant clades. This is a special case of sequential binary partitioning (SBP) (Egozcue & Pawlowsky-Glahn, 2005), which Silverman et al. (2017) applied to phylogenies as *PhILR*. PhILR is well-matched to that setting, as phylogenies are typically inferred as bifurcating, and here, orthonormality is straightforward: contrasts at disjoint nodes have disjoint support, and contrasts at nested nodes are orthogonal because the inner contrast sums to zero on each child clade. This construction works because binary branching imposes minimal local structure: each node requires exactly one contrast, yielding a one-to-one correspondence between internal nodes and basis vectors. The global consistency problem asking that local contrasts compose into an orthonormal basis on leaves reduces to verifying *pairwise* orthogonality, which holds by support structure and zero-sum constraints.

**What happens with polytomies?** For general trees $\mathcal{T}$, the simplicity above breaks down (Figure 1). Consider a node $u$ with $k_u > 2$ children. Comparing $k_u$ clades requires $k_u - 1$ orthogonal contrasts—a subspace, not a single vector. Several challenges arise: (i) defining canonical local contrasts at $u$, (ii) extending them to global vectors on leaves, and (iii) ensuring orthogonality across all nodes. Standard approaches fail because subtrees of different sizes contribute unequally to inner products, as detailed shortly.

**Polytomies are common in practice.** Polytomies arise within phylogenies as both hard polytomies (e.g., rapid radiations) and soft polytomies (e.g., collapsed low-support nodes) (Maddison, 1989). About 64% of taxonomic branch points in the NCBI Taxonomy Database have three or more children (Lin et al., 2011) and biomedical ontologies routinely encode multi-way groupings, e.g., the cell ontology (Diehl et al., 2016). Such curated trees often carry *meaningful internal structure* (e.g., independently named internal nodes) that can inform downstream representations. However, in practice, one typically *arbitrarily refines* them into binary trees (Lin et al., 2011). This introduces additional internal nodes and splits *not present in the original hierarchy*, making resulting coordinates and interpretations binarization-dependent. Alternative approaches aim to identify predictive log-ratio features via log-contrast regression, greedy balance selection (Rivera-Pinto et al., 2018), pair-

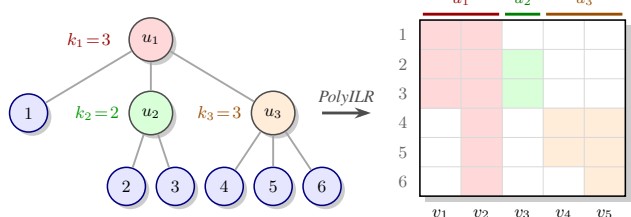

*Figure 2.* *PolyILR* from $\mathcal{T}$ on $d = 6$ leaves to $V \in \mathbb{R}^{d \times (d-1)}$. Colors indicate generating nodes; white entries are zeros.

wise log-ratio testing (Mandal et al., 2015), or phylogeny-guided ILR factors via edge selection (Washburne et al., 2017). But these methods yield isolated contrasts rather than a full, node-grouped orthonormal coordinate system canonically tied to a given multifurcating tree.

No existing method provides a canonical, complete orthonormal decomposition for general trees that simultaneously defines local contrasts, extends them globally, and ensures consistency. Obtaining such a decomposition without sacrificing isometry or introducing arbitrary choices is our goal.

# 4. PolyILR

We describe the high-level idea in §4.1, the formal construction in §4.2, and properties in §4.3.

## 4.1. From Hierarchy to Geometry

**Goal.** We seek an orthonormal decomposition of $\mathcal{H}$ reflecting the full branching structure of $\mathcal{T}$. We do not just want predictive log-ratios or useful balances, but a complete coordinate system where each coordinate corresponds to a location in $\mathcal{T}$. Such a basis must capture the full local structure at each node, maintain orthonormality within and across nodes, span $\mathcal{H}$, and be canonical. The first ensures complete encoding, the next two define a valid ILR basis in (3). And the last one ensures reproducibility. It is not obvious such a construction exists.

**Key insight.** To address this, we (i) associate local geometric structure to each internal node and (ii) assemble them into global structure. We attach to each node a structured object encoding all relative comparisons among its children. The specific structure and canonicity follow from requiring the global basis to be a valid ILR basis and a deterministic choice of local basis. We make this precise in §4.2.

**(i) Local structure.** Consider an internal node $u$ with $k_u$ children. The global basis will act on compositions, comparing leaves (via geometric mean) within each child clade. Since compositions carry only relative information (as in §2.1), we encode relative differences among these $k_u$ clades and not absolute levels. This requires $k_u - 1$ degrees of freedom: one contrast distinguishes two children, two contrasts distinguish three, and so on. Each internal node thus

contributes a $(k_u - 1)$-dimensional structure.

**(ii) Global assembly.** Local contrasts live at nodes, but ILR coordinates must be global vectors on leaves. We write a weighted inner product at each node to account for the number of descendant leaves in each child clade. Orthonormality under this weighted inner product (locally) guarantees global orthonormality in $\mathbb{R}^d$ after spreading to leaves.

We now formalize our construction, **PolyILR** (Figure 2).

### 4.2. PolyILR Construction

**Setup.** Let $\mathcal{T}$ be any rooted tree with $d$ leaves. Our data live in the Aitchison simplex $\Delta^{d-1}$, where each component corresponds to a leaf of $\mathcal{T}$. Our goal is to construct a valid ILR basis $V \in \mathbb{R}^{d \times (d-1)}$ such that each column of $V$ corresponds to a specific internal node of $\mathcal{T}$.

**Local contrast subspace.** Consider a node $u$ with $k_u$ children. We define the local contrast subspace at $u$ as

$$S_u = \left\{ h \in \mathbb{R}^{k_u} : \sum_{r=1}^{k_u} h_r = 0 \right\}. \tag{4}$$

This is the $(k_u - 1)$-dimensional subspace orthogonal to $\mathbf{1}$, capturing all relative comparisons among the $k_u$ children (see §4.1). Notice that this zero-sum constraint is not arbitrary: it is, in fact, forced by the ILR requirement that $V^\top \mathbf{1} = 0$. Since each column of $V$ must be orthogonal to $\mathbf{1}$, and each column is formed by spreading a local vector $\mathbf{h}$ from node $u$, we require $\mathbf{h}^\top \mathbf{1} = 0$ locally.

**Weighted inner product.** To ensure that local orthonormality extends globally, we equip $S_u$ with a *weighted inner product*. Let $n_r$ denote the number of leaves descending from child $r \in \{1, \ldots, k_u\}$. We define:

$$\langle h, h' \rangle_w = \sum_{r=1}^{k_u} \frac{h_r h'_r}{n_r}. \tag{5}$$

This accounts for unequal subtree sizes: children with more descendants contribute less per leaf to the global inner product when spread. Note that $(S_u, \langle \cdot, \cdot \rangle_w)$ is a Hilbert space.

**Local basis.** We choose an orthonormal basis of $S_u$. Any orthonormal basis works mathematically, but we use Helmert contrasts (Lancaster, 1965) for canonicity. The standard Helmert matrix $H \in \mathbb{R}^{k \times (k-1)}$ has columns:

$$H_{r,m} = \begin{cases} \sqrt{\frac{1}{m(m+1)}} & \text{if } r \leq m, \\ -\sqrt{\frac{m}{m+1}} & \text{if } r = m+1, \\ 0 & \text{if } r > m+1. \end{cases} \tag{6}$$

The $m$-th column compares child $m+1$ against the average

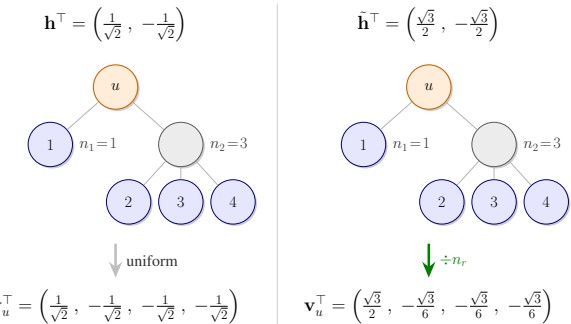

*Figure 3. Naive vs. weighted spreading (Example 4.1).* Uniform spreading (Left) vs. Weighted spreading (Right)

of children $1, \ldots, m$. For example, with $k_u = 3$ children:

$$H^{(u)} = \begin{pmatrix} \frac{1}{\sqrt{2}} & \frac{1}{\sqrt{6}} \\ -\frac{1}{\sqrt{2}} & \frac{1}{\sqrt{6}} \\ 0 & -\frac{2}{\sqrt{6}} \end{pmatrix}.$$

The first column contrasts child 2 versus child 1, the second contrasts child 3 versus the average of children 1 and 2. Helmert contrasts provide a canonical choice: given an ordering of children, the basis is deterministic. Alternative orthonormal bases of $S_u$ (e.g., QR decomposition) can also work but lack this sequential interpretability.

Here, the columns of $H^{(u)}$ are orthonormal under the standard inner product and lie in $S_u$. To obtain orthonormality under $\langle \cdot, \cdot \rangle_w$, we apply Gram-Schmidt to $H^{(u)}$ under this weighted inner product, yielding $\widetilde{H}^{(u)} \in \mathbb{R}^{k_u \times (k_u - 1)}$. This choice ensures that given $\mathcal{T}$ and a fixed ordering of children at each node, the local basis is *uniquely* obtained.

**Spreading to leaves.** Each column of the local basis $\widetilde{H}^{(u)}$ is a vector in $\mathbb{R}^{k_u}$, defined on the children of $u$. We spread it to a global vector $v \in \mathbb{R}^d$ on all leaves:

$$v_i = \begin{cases} \widetilde{H}^{(u)}_{r,m} / n_r & \text{if leaf } i \text{ descends from child } r, \\ 0 & \text{otherwise.} \end{cases}$$

Division by $n_r$ is key, as it ensures that orthonormality under $\langle \cdot, \cdot \rangle_w$ at node $u$ implies orthonormality in $\mathbb{R}^d$ after spreading. Consider two local vectors $\mathbf{h}, \mathbf{h}'$ orthonormal under $\langle \cdot, \cdot \rangle_w$: $\sum_r h_r h'_r / n_r = \delta_{\mathbf{h}, \mathbf{h}'}$. After spreading with the $1/n_r$ weighting, their global inner product becomes:

$$\langle \mathbf{v}, \mathbf{v}' \rangle = \sum_{i=1}^{d} v_i v'_i = \sum_{r=1}^{k_u} \sum_{i \in C_r^{(u)}} \left( \frac{h_r}{n_r} \right) \left( \frac{h'_r}{n_r} \right)$$

$$= \sum_{r=1}^{k_u} \frac{h_r h'_r}{n_r^2} \cdot n_r = \sum_{r=1}^{k_u} \frac{h_r h'_r}{n_r} = \delta_{\mathbf{h}, \mathbf{h}'},$$

where the second equality holds because child $r$ contributes exactly $n_r$ leaves, each with coefficient $h_r / n_r$ (similarly for $\mathbf{h}'$). So, local orthonormality means global orthonormality.

**Algorithm 1** PolyILR Basis Construction

**Require:** Rooted $T$ with $d$ leaves, internal node ordering $\pi$ (DFS)
**Ensure:** ILR basis $V \in \mathbb{R}^{d \times (d-1)}$
1: $j \leftarrow 1$
2: **for** each internal node $u$ from $\pi$ **do**
3:     $k_u \leftarrow$ number of children of $u$
4:     **for** $r = 1, \dots, k_u$ **do**
5:         $C_u^{(r)} \leftarrow$ leaves descending from $r$-th child
6:         $n_r \leftarrow |C_u^{(r)}|$
7:     **end for**
8:     $\mathcal{S}_u \leftarrow \{\mathbf{h} \in \mathbb{R}^{k_u} : \sum_r h_r = 0\}$
9:     $\langle \mathbf{h}, \mathbf{h}' \rangle_w \leftarrow \sum_r h_r h_r' / n_r$
10:    $H^{(u)} \leftarrow$ Helmert matrix in $\mathbb{R}^{k_u \times (k_u - 1)}$
11:    $\widetilde{H}^{(u)} \leftarrow$ Gram-Schmidt on $H^{(u)}$ under $\langle \cdot, \cdot \rangle_w$
12:    **for** $m = 1, \dots, k_u - 1$ **do**
13:       **for** $i = 1, \dots, d$ **do**
14:          **if** $i \in C_u^{(r)}$ for some $r$ **then**
15:             $V_{i,j} \leftarrow \widetilde{H}_{r,m}^{(u)} / n_r$
16:          **else**
17:             $V_{i,j} \leftarrow 0$
18:          **end if**
19:       **end for**
20:       $j \leftarrow j + 1$
21:    **end for**
22: **end for**
23: **return** $V$

*Example* 4.1. Consider $u$ with two children $n_1 = 1$ and $n_2 = 3$. The Helmert contrast is $\mathbf{h} = (1/\sqrt{2}, -1/\sqrt{2})^\top$, which satisfies $\|\mathbf{h}\|_2^2 = 1$, but spreading uniformly gives $\mathbf{v} = (1/\sqrt{2}, -1/\sqrt{2}, -1/\sqrt{2}, -1/\sqrt{2})^\top$ with $\|\mathbf{v}\|^2 = 2 \neq 1$. The weighted norm in (5) gives $\|\mathbf{h}\|_w^2 = (1/2)/1 + (1/2)/3 = 2/3$; normalizing gives $\tilde{\mathbf{h}} = [\sqrt{3}/2, -\sqrt{3}/2]^\top$. Spreading with $\div n_r$ yields $\|\mathbf{v}\|^2 = 1$. See Figure 3.

**Assembling it all.** Applying this procedure at every internal node of $\mathcal{T}$ and collecting all spread vectors yields the PolyILR basis $V$. The columns of $V$ are indexed by pairs $(u, m)$ for some internal node $u$ and a contrast index $m \in \{1, \dots, k_u - 1\}$. For example, if $\mathcal{T}$ has $\ell$ internal nodes ordered as $u_1, \dots, u_\ell$, with $u_1$ having 4 children, $u_2$ having 3, $\dots$, and $u_\ell$ having 3 children, then the basis is

$$V = \big(\underbrace{v_1, v_2, v_3}_{\text{node } u_1}, \underbrace{v_4, v_5}_{\text{node } u_2}, \dots, \underbrace{v_{d-2}, v_{d-1}}_{\text{node } u_\ell}\big).$$

See Algorithm 1 and Figure 2.

**Theorem 4.1** (PolyILR). *Let $\mathcal{T}$ be any rooted tree with $d$ leaves. The matrix $V \in \mathbb{R}^{d \times (d-1)}$ from Alg. 1 satisfies:*

   *1. $V^\top \mathbf{1} = 0$ (contrast property),*

   *2. $V^\top V = I_{d-1}$ (orthonormality).*

*Consequently, $\varphi(x) = V^\top \log x$ is an isometry from $(\Delta^{d-1}, \langle \cdot, \cdot \rangle_A)$ to $(\mathbb{R}^{d-1}, \langle \cdot, \cdot \rangle_2)$.*

*Proof idea.* The contrast property follows from each local contrast summing to zero. For orthonormality: vectors from

the same node are orthonormal by construction; disjoint nodes have disjoint support. Nested nodes (one ancestor of the other) are orthogonal because the descendant's spread vector sums to zero on each child clade. Full proof and algorithm details in Appendix A.

**Interpretation.** Given composition $x$, each coordinate $z_j = v_j^\top \log x$ is a *balance*: a log-ratio comparing geometric means of child clades at an internal node. The transformation $z = V^\top \log x \in \mathbb{R}^{d-1}$ decomposes $x$ into interpretable contrasts at every level of the hierarchy, see §5.

### 4.3. Properties of PolyILR

**Relation to existing methods.** When $\mathcal{T}$ is binary ($k_u = 2$ for all $u$), each node contributes *one* contrast, and PolyILR reduces to PhILR (see Appendix A). Unlike greedy balance selection (Rivera-Pinto et al., 2018) or edge-based factorization, which yield isolated contrasts, PolyILR provides a complete orthonormal basis aligned with the full tree. *Unlike arbitrary binarization, PolyILR respects the original topology without introducing artificial splits* (Figure 1).

**Uniqueness and recoverability.** PolyILR provides a *canonical* basis $V$ aligned with $\mathcal{T}$ as follows.

**Proposition 4.2.** *Given a rooted tree $\mathcal{T}$ with fixed leaf labels and child orderings, PolyILR produces a unique basis $V(\mathcal{T})$. Moreover, $\mathcal{T} \mapsto V(\mathcal{T})$ is injective: $\mathcal{T}$ can be recovered from $V$ via the clade supports of its columns.*

We point out that PolyILR's *canonicity* rests on two structural conventions: (i) a child ordering at each internal node and (ii) a sign convention for the Helmert columns (first nonzero entry positive). These fix the representation but not the underlying geometry: different orderings yield bases related by an orthogonal transformation within each node's block (and permutations across blocks) where sign flips change the orientation of contrasts. In practice, the ordering is inherited from the input tree and held fixed. Full proofs are in Appendix A.

*Summary.* PolyILR provides a canonical, tree-aligned orthonormal coordinate system for the Aitchison simplex. Each coordinate corresponds to a contrast at a specific internal node, enabling interpretable analysis at any resolution.

## 5. Structured Analysis with PolyILR

We describe how PolyILR coordinates enable structured analysis beyond what standard log-ratio transforms provide. Let $x \in \Delta^{d-1}$ be a composition with components as leaves of $\mathcal{T}$. The PolyILR transform yields $z = \varphi(x) \in \mathbb{R}^{d-1}$.

### 5.1. Tree-Aligned Coordinates

**Coordinate indexing.** By construction, each coordinate index $j$ corresponds bijectively to a pair $(u, m)$: an internal

node $u$ and a contrast index $m \in \{1, \ldots, k_u - 1\}$. The coordinate $z_j = z_{(u,m)}$ is a *balance*—a log-ratio comparing the geometric mean of leaves under child $m + 1$ against that under children $1, \ldots, m$ at node $u$ (see (6)). This association is intrinsic.

**Multiscale structure.** By construction, each internal node $u$ contributes $k_u - 1$ coordinates to the basis $V$ (see §4.2). Because these coordinates are orthonormal, the coordinates at distinct nodes span orthogonal subspaces, yielding a disjoint partition of $\mathbb{R}^{d-1}$ indexed by tree nodes. We can thus reason about node $u$ as a unit: do the coordinates at $u$ jointly explain an outcome? Does variation concentrate at $u$?

This node-level partition extends to coarser groupings. Aggregating nodes by tree depth or by subtree membership yields alternative orthogonal partitions of the same space (Table 1). PolyILR inherits them directly from the tree. We illustrate these partitions in Figure 5 (Appendix B).

| Aggregation | Question Answered |
|---|---|
| Node | Which *splits* drive signal? |
| Depth | What *resolution* matters? |
| Subtree | Is an entire *clade* informative? |

*Table 1.* Tree substructure aggregations enabled by PolyILR.

### 5.2. Implications for Inference

Given data $\{(x_i, y_i)\}_{i=1}^N$ with outcome $y_i$, we transform $z_i = \varphi(x_i)$ and fit any model on $\{(z_i, y_i)\}$. Since $\varphi$ is an isometry, the full geometry is preserved.

**Feature selection.** Identifying *which features drive outcomes* is a key scientific goal. In genomics, neuroscience, and microbiome alike, the goal is often not just prediction but understanding which variables matter and why (Rudin, 2019; Marcos-Zambrano et al., 2021). With standard ILR, important coordinates are anonymous indices with no semantics. With PolyILR, when coordinate $j = (u, m)$ is identified as important, we know which node and contrast drive the signal: a log-ratio comparing specific groups of leaves. This interpretability is intrinsic to the representation, no post-hoc processing needed.

**Tree-level aggregation.** The multiscale structure of Poly-ILR enables inference at any tree substructure. Let $\omega_j$ denote importance of coordinate $j$ from any method (e.g., random forest). We aggregate per-coordinate importances over any disjoint set $S$ of coordinates (node, depth, or subtree) via $\omega(S) = \sum_{j \in S} \omega_j$. This aggregation is well-defined because coordinates at distinct nodes span orthogonal subspaces, any partition of the tree into disjoint substructures yields a partition of $\mathbb{R}^{d-1}$, and the corresponding importances sum to the total without double-counting.

**Leaf-level importance.** To quantify importance of an individual leaf $\ell$ (e.g., a taxon), we cannot directly aggregate coordinates because coordinates are not exclusive to any single leaf. Instead, we can distribute importance weighted by participation. By construction, $V_{\ell j}$ quantifies how much leaf $\ell$ participates in coordinate $j$. We define $\omega(\ell) = \sum_{j=1}^{d-1} V_{\ell j}^2 \cdot \omega_j$. Since columns of $V$ are unit vectors, $\sum_\ell V_{\ell j}^2 = 1$, so leaf importances sum to total importance.

| Task | CLR | | | PhILR | | | PolyILR | | |
|---|---|---|---|---|---|---|---|---|---|
| | RF / SVM / LR | | | RF / SVM / LR | | | RF / SVM / LR | | |
| *Acc (%)* | | | | | | | | | |
| **HMP** body sites (5) | .956/.971/.962 | | | .961/.971/.962 | | | .963/.971/.962 | | |
| body subsites (18) | .597/.672/.646 | | | .608/.672/.646 | | | .622/.672/.646 | | |
| **cMD3** westernized (2) | .972/.979/.968 | | | .966/.979/.967 | | | .967/.979/.967 | | |
| age category (5) | .785/.814/.739 | | | .797/.814/.738 | | | .797/.814/.738 | | |
| **DISCO** leukemia (2) | .925/.932/.927 | | | .940/.932/.927 | | | .935/.932/.927 | | |
| HCC (2) | .921/.927/.944 | | | .910/.927/.944 | | | .910/.927/.944 | | |
| *AUROC* | | | | | | | | | |
| **HMP** body sites (5) | .987/.995/.994 | | | .992/.995/.994 | | | .992/.995/.994 | | |
| body subsites (18) | .957/.974/.967 | | | .965/.974/.967 | | | .966/.974/.967 | | |
| **cMD3** westernized (2) | .975/.978/.967 | | | .966/.978/.966 | | | .966/.978/.967 | | |
| age category (5) | .836/.867/.833 | | | .837/.867/.833 | | | .843/.867/.833 | | |
| **DISCO** leukemia (2) | .968/.984/.982 | | | .984/.984/.982 | | | .984/.984/.982 | | |
| HCC (2) | .921/.996/.998 | | | .984/.996/.998 | | | .974/.996/.998 | | |

| Task | PolyILR | | | PhILR (index / semantic) | | |
|---|---|---|---|---|---|---|
| | $K$=5 | $K$=10 | $K$=50 | $K$=5 | $K$=10 | $K$=50 |
| **HMP** body sites (5) | .66 | .65 | .84 | .01 / .08 | .01 / .06 | .07 / .05 |
| body subsites (18) | .71 | .72 | .88 | .01 / .22 | .02 / .13 | .07 / .04 |
| **cMD3** westernized (2) | .43 | .69 | .81 | .00 / .01 | .00 / .01 | .01 / .02 |
| age category (5) | .73 | .58 | .76 | .12 / .04 | .09 / .03 | .03 / .03 |
| healthy vs disease (2) | .56 | .86 | .80 | .00 / .02 | .00 / .03 | .02 / .02 |
| **DISCO** leukemia (2) | .75 | .81 | .92 | .05 / .09 | .12 / .16 | .73 / .12 |
| HCC (2) | .77 | .78 | .84 | .03 / .07 | .06 / .09 | .35 / .11 |

*Table 2.* (**Top**) *Classification accuracy and AUROC* (5 runs) across CLR, PhILR, and PolyILR. Each cell reports RF/SVM/LR. SVM and LR match across the three ILR representations within each task, as expected from isometry; differences are confined to RF but modest. Full statistical variability (95% CIs) is in Appendix B.7 but observed to be small. (**Bottom**) *Feature stability* (Jaccard of top-$K$ features across CV folds). PolyILR stable; PhILR (index/semantic) unstable from arbitrary binarization.

## 6. Experiments

We evaluate PolyILR on standard microbiome and single-cell benchmarks. Our goals are to demonstrate PolyILR provides: (**G1**) valid ILR representations (§6.2), (**G2**) stable feature selection, unlike PhILR with arbitrary binarization (§6.3), (**G3**) interpretable features grounded in the tree (§6.4), and (**G4**) structured inference by tree subparts (§6.5).

### 6.1. Setup

**Datasets.** We use three large datasets from two domains. For microbiome: HMP (Human Microbiome Project; 4,743 samples, 402 taxa) (Human Microbiome Project Consortium, 2012) and cMD3 (curatedMetagenomicData v3;

| | Rank | Contrast | RF Importance (%) | Rank range |
|---|---|---|---|---|
| HMP body sites | 1 | Streptococcaceae vs Lactobacillus + Leuconostocaceae | 3.94 | 1 |
| | 2 | Lactococcus vs Streptococcus | 3.42 | 2–3 |
| | 3 | Pseudomonadales vs Cardiobacteriaceae + Vibrionaceae + Legionellales + ... | 3.40 | 2–3 |
| | 4 | Bacillales vs Gemella + Exiguobacterium + Turicibacter + ... | 2.95 | 4–5 |
| cMD 3 westernized | 1 | Prevotella vs Alistipes + Anaerotruncus + Bacteroides + ... | 1.77 | 1 |
| | 2 | Murimonas vs Alistipes + Anaerotruncus + Bacteroides + ... | 1.49 | 2–4 |
| | 3 | Prevotella vs Bacteroides + Alistipes | 1.45 | 2–5 |
| | 4 | Lactobacillus vs Alistipes + Anaerotruncus + Bacteroides + ... | 1.38 | 4–6 |
| cMD 3 health/dis. | 1 | Lachnoclostridium vs Bacteroides + Turicibacter + Ruminococcus + ... | 0.55 | 1 |
| | 2 | Bifidobacterium vs Blautia + Enterococcus | 0.38 | 2–5 |
| | 3 | Flavonifractor vs Bifidobacterium + Actinomyces | 0.35 | 2–6 |
| | 4 | Fusicatenibacter vs Gemmiger | 0.34 | 2–6 |
| DISCO leukemia | 1 | Myeloid cell vs Erythrocyte/Megakaryocyte + Hematopoietic precursor cell | 14.50 | 1 |
| | 2 | Cycling T/NK cell vs ILC + T cell + NK cell | 11.79 | 2 |
| | 3 | Lymphoid cell vs Erythrocyte/Megakaryocyte + Hematopoietic precursor + Myeloid cell | 6.30 | 3–4 |
| | 4 | MAIT cell vs Naive T cell + Memory T cell + CD8 T cell + ... | 5.45 | 3–5 |
| DISCO HCC | 1 | Erythrocyte/Megakaryocyte vs Myeloid cell + Lymphoid cell + Hematopoietic precursor cell | 10.08 | 1 |
| | 2 | B cell precursor vs Plasma cell + INF-activated naive B cell + Memory B cell + ... | 6.52 | 2–4 |
| | 3 | Hematopoietic precursor cell vs Myeloid cell + Lymphoid cell | 6.34 | 2–3 |
| | 4 | Venous EC vs LSEC | 5.45 | 2–8 |

*Table 3.* Top-4 PolyILR contrasts by RF importance (%), with rank range across 5-fold CV. Each contrast is a coordinate representing the log-ratio of geometric means between two groups. "A vs B + C" means the log-ratio of geometric means of A's descendants against the pooled descendants of B and C; "+ ..." marks additional siblings omitted. Full importance variability (mean ± std) is in Appendix B.7.

20,238 samples from 86 studies, 2,047 taxa) (Pasolli et al., 2017), with taxonomies from NCBI. For single-cell biology: DISCO (Database of Immune Single-Cell Omics; 751 sample-level composition profiles derived from ∼5.3M cells, 62–99 cell types) (Li et al., 2022), with cell types organized by the Cell Ontology. Tasks include predictions on body site (5–18 sites), westernization, age category, healthy vs. disease (microbiome), and healthy vs. leukemia/HCC (single-cell). As with any log-ratio method, PolyILR requires zero handling before the coordinate transform; we use a small additive pseudocount per dataset (Appendix B.4).

**Methods.** We compare PolyILR against CLR (same geometry, no tree alignment) and unweighted PhILR with random binarization (tree-aligned, but defined on binary trees, so polytomies must be resolved arbitrarily). We use random forest (RF), SVM, and logistic regression (LR) with 5-fold cross-validation. Additional experiments, hyperparameter, and dataset construction details are in Appendix B.

*Remark* 6.1. PolyILR is a representation (coordinate system), **not** a task-specific model. So, *any* downstream analysis may be applied to the resulting coordinates. Conclusions and scientific validity depend on appropriate statistical methodology.

### 6.2. Representation Validity

We verify PolyILR is a geometrically valid representation **(G1)**. CLR projects compositions into the tangent space $\mathcal{H}$, spanned by PolyILR coordinates (Thm. 4.1). Table 2 (top) supports this equivalence: CLR, PhILR, and PolyILR yield identical SVM and LR accuracy, as expected from isometry. RF accuracy varies modestly across representations (within ±1.5% on most tasks), since RF is sensitive to the choice of

axes; differences in either direction are consistent with the geometric equivalence.

### 6.3. Feature selection is stable

A key advantage of PolyILR over PhILR is *canonical* decomposition on any tree **(G2)**. PhILR requires binarizing polytomies, making feature selection unstable across binarizations with *no correct choice*. We measure stability via: (i) *index stability* (Jaccard similarity of top-$K$ feature indices across runs) measuring if the same coordinate positions are selected; and (ii) *semantic stability* (similarity of the corresponding taxonomic/ontological contrasts) measuring whether selected features represent the same biological comparisons regardless of index. The latter is fairer to PhILR as it ignores arbitrary index assignment. For PhILR, we vary the random binarization of the same polytomous tree across runs; for PolyILR, (i) and (ii) coincide since coordinates are canonical given the tree. Table 2 (bottom) shows PolyILR achieves high stability (0.43–0.92) while PhILR collapses (near 0) under both metrics across all three datasets. Even when comparing semantically, PhILR's artificial binary splits yield different partitions across binarizations, confirming that the instability is structural (Figure 1).

### 6.4. Features are interpretable

PolyILR coordinates are directly interpretable as taxonomic/ontological contrasts **(G3)**. Table 3 shows the top-4 features by RF importance, with rank range across 5 runs indicating stability of the ranking. Each feature is a log-ratio contrast between groups at a specific node (§5.2).

*Scientific interpretation (Table 3).* The recovered contrasts are consistent with previously reported observations in the

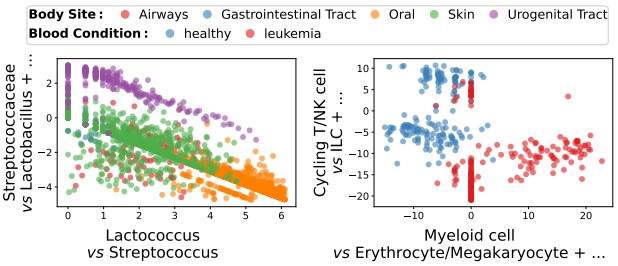

**Body Site :** ● Airways ● Gastrointestinal Tract ● Oral ● Skin ● Urogenital Tract
**Blood Condition :** ● healthy ● leukemia

*Figure 4.* Projection onto top-2 PolyILR features for `HMP` body site classification (left) and `DISCO` leukemia classification (right). Axes are the two most predictive contrasts (Table 3).

| Structure | body sites (5) | | | body subsites (18) | | |
|---|---|---|---|---|---|---|
| | Component | Acc | Imp. | Component | Acc | Imp. |
| Depth | ≤0 (coarsest) | .87 | 6.8% | ≤0 (coarsest) | .42 | 4.8% |
| | ≤2 | .96 | 47% | ≤2 | .59 | 44% |
| | ≤4 (all) | .96 | 100% | ≤4 (all) | .62 | 100% |
| Subtree | Firmicutes | .95 | 47% | Firmicutes | .55 | 36% |
| | Proteobacteria | .87 | 20% | Proteobacteria | .44 | 26% |
| | Actinobacteria | .91 | 19% | Actinobacteria | .45 | 22% |
| Node | Actinomycetales | – | 11% | Actinomycetales | – | 12% |
| | Lactobacillales | – | 9.1% | Lachnospiraceae | – | 8.4% |
| | Gammaproteobacteria | – | 8.0% | Clostridiales | – | 3.8% |
| Taxon | Streptococcus | – | 3.1% | Oribacterium | – | 1.0% |
| | Lactococcus | – | 3.1% | Corynebacterium | – | 1.0% |
| | Pasteurella | – | 2.6% | Lactococcus | – | 1.0% |

*Table 4.* Tree-level inference on `HMP`. Importance aggregated by depth (cumulative), subtree (phylum), node, and taxon.

| Structure | westernized (2) | | | healthy/disease (2) | | |
|---|---|---|---|---|---|---|
| | Component | Acc. | Imp. | Component | Acc. | Imp. |
| Depth | ≤0 (coarsest) | .934 | 0.1% | ≤0 (coarsest) | .613 | 0.3% |
| | ≤3 | .956 | 5.3% | ≤3 | .667 | 6.4% |
| | ≤6 (all) | .967 | 100% | ≤6 (all) | .682 | 100% |
| Taxon | Prevotella | – | 1.7% | Lachnoclostridium | – | 0.6% |
| | Murimonas | – | 1.4% | Bifidobacterium | – | 0.3% |
| | Bacteroides | – | 1.3% | Enterococcus | – | 0.3% |

*Table 5.* Tree-level inference on `cMD3`. Importance aggregated by depth (cumulative) and taxon.

| Structure | leukemia (2) | | | HCC (2) | | |
|---|---|---|---|---|---|---|
| | Component | Acc. | Imp. | Component | Acc. | Imp. |
| Depth | ≤0 (coarsest) | .62 | 4.6% | ≤0 (coarsest) | .88 | 8.1% |
| | ≤1 | .88 | 26% | ≤1 | .91 | 38% |
| | ≤6 (all) | .94 | 100% | ≤6 (all) | .91 | 100% |
| Subtree | Immune cell | .93 | 95% | Immune cell | .92 | 79% |
| | – | – | – | Endothelial cell | .80 | 7.8% |
| | – | – | – | Epithelial cell | .78 | 4.1% |
| Node | Immune cell | – | 22% | Immune cell | – | 19% |
| | T cell | – | 16% | Hema. precursor | – | 9.6% |
| | T/NK cell | – | 14% | Endothelial cell | – | 7.8% |
| Cell type | Cycling T/NK cell | – | 11.3% | GMP | – | 4.4% |
| | MAIT cell | – | 5.0% | Erythroblast (int.) | – | 4.2% |
| | Naive CD8 T cell | – | 4.1% | Erythroblast (late) | – | 4.2% |

*Table 6.* Tree-level inference on `DISCO`. Importance aggregated by depth (cumulative), subtree, node, and cell type.

literature. For HMP body sites, Streptococcus vs. Lactococcus (3.4%) reflects known niche specialization within Streptococcaceae across oral subsites (Human Microbiome Project Consortium, 2012; Dewhirst et al., 2010), with Lactococcus lactis reported as a prevalent lactic-acid bacterium in the gut (Pasolli et al., 2020). For westernization, Prevotella vs. Bacteroides/Alistipes (1.5–1.8%) captures the lifestyle axis, with Prevotella enriched in non-Western populations consuming plant-rich diets (De Filippo et al., 2010; Yatsunenko et al., 2012). For healthy vs. disease, Lachnoclostridium (0.6%) aligns with documented links to colorectal cancer and atherosclerosis (Cai et al., 2022; Liang et al., 2020). For leukemia, myeloid vs. erythroid/precursor imbalance (14.5%) reflects lineage disruption in hematological malignancies (Löwenberg et al., 1999). For HCC, venous EC vs. LSEC (5.5%) captures the well-documented dedifferentiation of liver sinusoidal endothelial cells in hepatocellular carcinoma (Sørensen et al., 2015).

*Geometric structure.* Figure 4 projects `HMP` samples onto the top-2 coordinates. Unlike PCA, *each axis here is a single interpretable contrast* rather than a linear combination of all features. We do not claim maximal variance explained; rather, biologically meaningful features alone suffice to separate body sites. The linear substructures within classes may reflect shared sparsity: samples with identical zero-count taxa map to parallel manifolds in ILR space.

### 6.5. Tree-Level Inference

PolyILR enables structured hypothesis testing at multiple resolutions **(G4)**. RF importance can be aggregated by depth, subtree, node, or leaf (see §5.2). We report all four levels for `HMP` (Table 4) and `DISCO` (Table 6), but only depth and taxon for `cMD3` (Table 5) whose meta-analytic tree lacks consistent intermediate labels. Subtree-level partitions by root's children (root omitted).

*Scientific interpretation (Tables 4–6).* Aggregations agree with known structure. For HMP, Firmicutes (47%) and Proteobacteria (19%) dominate body site signals (Human Microbiome Project Consortium, 2012; Costello et al., 2009; Ma et al., 2024). At node level, Actinomycetales (11%) and Lactobacillales (9%) capture skin vs. oral distinctions. For cMD3 westernization, coarse contrasts (≤3) achieve 95.6% acc., consistent with diet-associated shifts at coarse resolution (De Filippo et al., 2010; Arumugam et al., 2011). For DISCO leukemia (single-cell), 95% of importance concentrates in Immune cells, with T cell (16%) and T/NK cell (14%) nodes dominating (Löwenberg et al., 1999). For HCC, importance distributes across Immune (79%), Endothelial (8%), and Epithelial (4%) subtrees, reflecting multi-compartment remodeling (Sørensen et al., 2015).

We should note that the biological interpretations above are *plausibility checks consistent with prior literature.* Any causal or clinical conclusions will require much deeper analyses beyond the scope of this methodological work.

In summary, PolyILR addresses all goals **(G1–G4)** while recovering features consistent with known biomarkers.

# 7. Beyond Compositional Data

We establish a connection between compositional data and probabilistic modeling via shared underlying geometry. Further analysis and validation may be of independent interest.

**Aitchison geometry as quotient.** Compositional data identifies vectors up to equivalence classes $[\mathbf{c}]$ induced by $\mathbf{c} \sim_c \lambda\mathbf{c}$ for $\lambda > 0$ (i.e., scaling), since only ratios carry information. We observe that the quotient $\mathbb{R}^d_{>0}/\sim_c$ is the Aitchison simplex, whose tangent space is $\mathcal{H}$ via CLR (Aitchison, 1982; Egozcue et al., 2003).

**Probabilistic modeling.** Consider a model $f_\theta$ outputting logits $\mathbf{z} = f_\theta(\mathbf{x}) \in \mathbb{R}^d$, with predicted distribution $\mathbf{p} = \mathrm{softmax}(\mathbf{z})$ trained via cross-entropy. Since softmax is *shift-invariant*, i.e., $\mathrm{softmax}(\mathbf{z} + c\mathbf{1}) = \mathrm{softmax}(\mathbf{z})$, this induces an equivalence relation $\mathbf{z} \sim_\ell \mathbf{z} + c\mathbf{1}$. The loss and predictions are invariant to shifts along $\mathbf{1}$.

**Proposition 7.1.** *Let $\mathcal{L} = \mathbb{R}^d/\sim_\ell$ be the quotient of logits under shift equivalence. Then $\mathcal{L} \cong \mathcal{H}$ (isomorphism). For $\mathbf{p} = \mathrm{softmax}(\mathbf{z})$, we have $\mathbf{z} - \bar{z}\mathbf{1} = \mathrm{clr}(\mathbf{p})$,    where $\bar{z} = \frac{1}{d}\sum_i z_i$, and $[\mathbf{z}] \to \mathbf{z} - \bar{z}\mathbf{1}$ is well-defined.*

The individual components (i.e., the CLR hyperplane, centering map (Egozcue et al., 2003), and softmax shift-invariance) are well-known. Proposition 7.1 newly establishes that after quotienting by the shift symmetry, logit space aligns with the CLR/Aitchison tangent space.

**Implications.** Many datasets have tree structure over classes, e.g., a superclass hierarchy for CIFAR-100 and WordNet for ImageNet (Deng et al., 2009; Miller, 1995). Given such a tree $\mathcal{T}$, PolyILR can transform model logits as $\mathbf{a} = \mathbf{V}^\top\mathbf{z}$ in tree-aligned coordinates where each component corresponds to a node and contrast. Since $\mathbf{p} = \mathrm{softmax}(\mathbf{Va})$, predictions can be analyzed in this interpretable space. For instance, the gradient w.r.t. logits is $\nabla_{\mathbf{z}}\ell = \mathbf{p} - \mathbf{e}_y$, with one-hot target $\mathbf{e}_y$. The gradient w.r.t. PolyILR coordinates thus is $\nabla_{\mathbf{a}}\ell = \mathbf{V}^\top\nabla_{\mathbf{z}}\ell = \mathbf{V}^\top(\mathbf{p} - \mathbf{e}_y)$. Each $(\nabla_{\mathbf{a}}\ell)_{(u,m)}$ measures how strongly the loss pushes probability mass along contrast $m$ at node $u$, localizing model errors in $\mathcal{T}$. See Appendix A and B.6 for proof and preliminary results. Practical applications to model training or analysis remain an open direction.

# 8. Related Work

**Compositional hierarchy methods.** ILR transform provides orthonormal coordinates for compositional data (Egozcue et al., 2003). PhILR (Silverman et al., 2017) aligns this transform with phylogenetic trees, the setting it was designed for, where binary topology is the standard convention; applying it to polytomous trees requires arbitrary binary resolution. UniFrac (Lozupone & Knight, 2005) incorporates phylogenetic information but produces a dissimilarity measure, not a coordinate system. Phylofactorization (Washburne et al., 2017) and selbal (Rivera-Pinto et al., 2018) target biomarker discovery and identify predictive balances via greedy selection, but yield task-specific contrasts rather than a complete basis. Dirichlet-tree models (Mao & Ma, 2022) use tree structure for clustering but operate probabilistically, not geometrically. PolyILR complements this line of work by providing a complete orthonormal decomposition for *any* tree. Other work compares proportion-based and compositional normalizations (Yerke et al., 2024).

**Trees and geometric representations.** A separate line of work studies geometric representations of trees themselves. Hyperbolic embeddings learn representations of hierarchical data in spaces of constant negative curvature (Nickel & Kiela, 2017; Chami et al., 2019; Sala et al., 2018), while tropical geometry and BHV tree space study geodesics and statistics over spaces *of* trees (Billera et al., 2001; Owen & Provan, 2010; Monod et al., 2018). These embed trees or treat them as data; PolyILR differs in that the tree is a fixed input that structures a decomposition of the data space.

# 9. Conclusion

We introduced PolyILR, a canonical orthonormal decomposition of the Aitchison simplex for arbitrary tree topologies, including polytomous ones common in real taxonomies and ontologies. The construction equips each internal node with a weighted local geometry that assembles into a global ILR basis, addressing the incompatibility between Aitchison geometry and polytomous hierarchies. Experiments on microbiome and single-cell data show stable, interpretable features with inference at multiple tree resolutions. A connection to softmax classifiers suggests applications to hierarchical probabilistic modeling.

**Limitations.** PolyILR requires a *known, fixed tree* as input and does not accommodate topological uncertainty (e.g., bootstrap support, posterior distributions over trees). We also know that external trees such as taxonomies and phylogenies may also contain noise. We provide a robustness analysis under nearest-neighbor interchange perturbations in Appendix B.5, and view extensions that propagate tree uncertainty into the coordinate system (e.g., support-weighted local geometries) as future work. PolyILR's coordinates are also only locally interpretable by construction, as those under different internal nodes live in distinct weighted subspaces and are not directly comparable, though cross-subtree summaries are recovered via tree-substructure aggregation in Section 5.2. Finally, like all log-ratio methods, PolyILR requires zero replacement as preprocessing (Section 6.1; Appendix B.4).

## Acknowledgments

We thank the anonymous reviewers for their constructive feedback. We are grateful to Prof. Colin Dewey and Prof. Christina Kendziorski for discussions and feedback. Authors were all partly supported by NIH R01AG092220.

## Impact Statement

This paper presents work whose goal is to advance the field of Machine Learning. There are many potential societal consequences of our work, none which we feel must be specifically highlighted here.

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

# A. Proofs and Algorithm Details

This section contains full proofs of the main theoretical results and detailed algorithm specifications.

## A.1. Proofs

**Theorem A.1** (Restatement of Theorem 4.1). *Let $\mathcal{T}$ be any rooted tree with $d$ leaves. The matrix $V \in \mathbb{R}^{d \times (d-1)}$ from Algorithm 1 satisfies:*

1. *$V^\top \mathbf{1} = 0$ (contrast property),*

2. *$V^\top V = I_{d-1}$ (orthonormality).*

*Consequently, $\varphi(x) = V^\top \log x$ is an isometry from $(\Delta^{d-1}, \langle \cdot, \cdot \rangle_A)$ to $(\mathbb{R}^{d-1}, \langle \cdot, \cdot \rangle_2)$.*

*Proof.* Let $T$ be a rooted tree with $d$ leaves. For each internal node $u$ with $k_u$ children, let $C_u^{(r)}$ denote the set of leaves descending from child $r$, and let $n_r = |C_u^{(r)}|$. The construction produces a local basis $\tilde{H}^{(u)} \in \mathbb{R}^{k_u \times (k_u-1)}$ that is orthonormal under the weighted inner product $\langle h, h' \rangle_w = \sum_{r=1}^{k_u} h_r h'_r / n_r$ and satisfies the contrast condition $\sum_{r=1}^{k_u} \tilde{H}_{r,m}^{(u)} = 0$ for all $m$. Each local vector is spread to a global vector $v^{(u,m)} \in \mathbb{R}^d$ via

$$v_i^{(u,m)} = \begin{cases} \tilde{H}_{r,m}^{(u)}/n_r & \text{if } i \in C_u^{(r)} \text{ for some } r \in \{1, \ldots, k_u\}, \\ 0 & \text{otherwise.} \end{cases}$$

The precise procedure is given in Algorithm 1. It suffices for us to show that $V$ (i) satisfies the contrast property (hence its columns span the tangent space $\mathcal{H}$), (ii) is orthonormal within each node, and (iii) is orthonormal across distinct nodes.

*(i) Contrast property.* For any internal node $u$ and contrast index $m$,

$$\sum_{i=1}^d v_i^{(u,m)} = \sum_{r=1}^{k_u} \sum_{i \in C_u^{(r)}} \frac{\tilde{H}_{r,m}^{(u)}}{n_r} = \sum_{r=1}^{k_u} n_r \cdot \frac{\tilde{H}_{r,m}^{(u)}}{n_r} = \sum_{r=1}^{k_u} \tilde{H}_{r,m}^{(u)} = 0,$$

where the last equality holds because $\tilde{H}^{(u)}$ is obtained by applying Gram-Schmidt to Helmert contrasts, which lie in the subspace $S_u = \{h \in \mathbb{R}^{k_u} : \sum_r h_r = 0\}$, and Gram-Schmidt preserves this subspace.

*(ii) Within-node orthonormality.* For any internal node $u$ and contrast indices $m_1, m_2 \in \{1, \ldots, k_u - 1\}$,

$$\langle v^{(u,m_1)}, v^{(u,m_2)} \rangle = \sum_{i=1}^d v_i^{(u,m_1)} v_i^{(u,m_2)} = \sum_{r=1}^{k_u} \sum_{i \in C_u^{(r)}} \frac{\tilde{H}_{r,m_1}^{(u)}}{n_r} \cdot \frac{\tilde{H}_{r,m_2}^{(u)}}{n_r}$$

$$= \sum_{r=1}^{k_u} n_r \cdot \frac{\tilde{H}_{r,m_1}^{(u)} \tilde{H}_{r,m_2}^{(u)}}{n_r^2} = \sum_{r=1}^{k_u} \frac{\tilde{H}_{r,m_1}^{(u)} \tilde{H}_{r,m_2}^{(u)}}{n_r}$$

$$= \langle \tilde{H}_{\cdot,m_1}^{(u)}, \tilde{H}_{\cdot,m_2}^{(u)} \rangle_w = \delta_{m_1,m_2},$$

where the last equality holds by construction of $\tilde{H}^{(u)}$ as an orthonormal basis under $\langle \cdot, \cdot \rangle_w$.

*(iii) Across-node orthogonality.* Consider two distinct internal nodes $u \neq u'$ with contrast indices $m$ and $m'$ respectively. We consider two cases. (a) If the subtrees rooted at $u$ and $u'$ share no leaves, then $v^{(u,m)}$ and $v^{(u',m')}$ have disjoint supports, so $\langle v^{(u,m)}, v^{(u',m')} \rangle = 0$. (b) Suppose without loss of generality that $u'$ is a descendant of $u$. Then $u'$ lies entirely within the subtree of exactly one child of $u$, say child $s$, so the support of $v^{(u',m')}$ is contained in $C_u^{(s)}$. On $C_u^{(s)}$, the vector $v^{(u,m)}$ takes the constant value $\tilde{H}_{s,m}^{(u)}/n_s$. Since $v_i^{(u',m')} = 0$ for $i \notin C_u^{(s)}$,

$$\langle v^{(u,m)}, v^{(u',m')} \rangle = \sum_{i \in C_u^{(s)}} v_i^{(u,m)} v_i^{(u',m')} = \frac{\tilde{H}_{s,m}^{(u)}}{n_s} \sum_{i \in C_u^{(s)}} v_i^{(u',m')} = \frac{\tilde{H}_{s,m}^{(u)}}{n_s} \cdot 0 = 0,$$

where the last equality uses (i): the entries of $v^{(u',m')}$ sum to zero, and all nonzero entries lie within $C_u^{(s)}$.

*Conclusion.* (i), (ii), and (iii) establish that $V^\top \mathbf{1} = 0$ and $V^\top V = I_{d-1}$. For the dimension count, let $N_{\text{int}}$ denote the number of internal nodes. Every non-root node has exactly one parent, so the number of edges satisfies $|E| = d + N_{\text{int}} - 1$. Since $|E| = \sum_u k_u$, we have

$$\sum_u (k_u - 1) = \sum_u k_u - N_{\text{int}} = (d + N_{\text{int}} - 1) - N_{\text{int}} = d - 1 = \dim(\mathcal{H}).$$

By standard ILR theory, any $V \in \mathbb{R}^{d \times (d-1)}$ satisfying $V^\top \mathbf{1} = 0$ and $V^\top V = I_{d-1}$ has columns forming an orthonormal basis of $\mathcal{H}$, the image of clr. Thus $\phi(x) = V^\top \log x$ is an isometry from $(\Delta^{d-1}, \langle \cdot, \cdot \rangle_A)$ to $(\mathbb{R}^{d-1}, \langle \cdot, \cdot \rangle_2)$. $\qquad \square$

**Proposition A.2** (Restatement of Proposition 4.2). *Given a rooted tree $\mathcal{T}$ with fixed leaf labels, fixed child orderings, and a fixed ordering $\pi$ of internal nodes (e.g., DFS), PolyILR produces a unique basis $V(\mathcal{T})$. Moreover, $\mathcal{T} \mapsto V(\mathcal{T})$ is injective: $\mathcal{T}$ can be recovered constructively from $V$.*

*Proof.* We prove uniqueness and recoverability separately.

*(i) Uniqueness.* The PolyILR construction (Algorithm 1) is deterministic given the tree $T$, leaf labels, child orderings at each internal node, and an ordering $\pi$ of internal nodes. At each node $u$ with $k_u$ children, the construction follows:

- The descendant sets $C_u^{(r)}$ and counts $n_r = |C_u^{(r)}|$ are determined by $T$ and the leaf labels, and the corresponding weighted inner product $\langle h, h' \rangle_w = \sum_r h_r h'_r / n_r$ is determined by $\{n_r\}$.

- The Helmert matrix $H^{(u)} \in \mathbb{R}^{k_u \times (k_u - 1)}$ is determined by $k_u$ and the child ordering. The weighted-orthonormal basis $\tilde{H}^{(u)}$ is obtained by applying Gram-Schmidt to $H^{(u)}$ under $\langle \cdot, \cdot \rangle_w$. Gram-Schmidt is made unique by normalizing each output vector to unit $w$-norm and adopting a fixed sign convention (e.g., the first nonzero entry in child order is positive). Gram-Schmidt works as the weighted inner product forms a Hilbert space over the subspace $\mathcal{S}_u$.

- The spreading operation is deterministic given $C_u^{(r)}$ and $n_r$.

Thus $V(T)$ is uniquely determined. This makes PolyILR *canonical*, i.e., the construction does not incur intractable randomness.

*(ii) Recoverability.* We show that the tree topology can be recovered from $V$.

*Support structure:* By the structure of Helmert contrasts, column $m$ of $H^{(u)}$ has nonzero entries only in rows $1, \ldots, m+1$. Since Gram-Schmidt orthonormalizes sequentially, $\tilde{H}_{\cdot,m}^{(u)}$ is a linear combination of $H_{\cdot,1}^{(u)}, \ldots, H_{\cdot,m}^{(u)}$, so $\tilde{H}_{r,m}^{(u)} = 0$ for $r > m+1$. Moreover, columns $1, \ldots, m-1$ of $H^{(u)}$ have zeros in row $m+1$, so orthogonalizing against them does not affect entry $(m+1, m)$; thus $\tilde{H}_{m+1,m}^{(u)} = H_{m+1,m}^{(u)} \neq 0$. After spreading, the support of $v^{(u,m)}$ is $\bigcup_{r=1}^{m+1} C_u^{(r)}$, and the supports at node $u$ form a *strictly nested chain*:

$$\text{supp}(v^{(u,1)}) \subsetneq \text{supp}(v^{(u,2)}) \subsetneq \cdots \subsetneq \text{supp}(v^{(u,k_u-1)}) = \bigcup_{r=1}^{k_u} C_u^{(r)}.$$

*Recovering the child clades of a fixed node.* Fix an internal node $u$ with children $C_u^{(1)}, \ldots, C_u^{(k_u)}$. From the support-structure argument above,

$$\text{supp}(v^{(u,m)}) = \bigcup_{r=1}^{m+1} C_u^{(r)}.$$

For $m = 2, \ldots, k_u - 1$, the child clades $C_u^{(m+1)} = \text{supp}(v^{(u,m)}) \setminus \text{supp}(v^{(u,m-1)})$ are determined by the support chain alone. It remains to recover $C_u^{(1)}$ and $C_u^{(2)}$ from $\text{supp}(v^{(u,1)}) = C_u^{(1)} \cup C_u^{(2)}$. Since there are no previous columns to orthogonalize against, the first weighted Gram-Schmidt output satisfies $\tilde{H}_{\cdot,1}^{(u)} = \alpha H_{\cdot,1}^{(u)}$ for some $\alpha > 0$. The first Helmert column has opposite signs in rows 1 and 2, so $\tilde{H}_{1,1}^{(u)} > 0$ and $\tilde{H}_{2,1}^{(u)} < 0$. After spreading, the values $\tilde{H}_{1,1}^{(u)}/n_1 > 0 > \tilde{H}_{2,1}^{(u)}/n_2$ are distinct, so the two level sets of $v^{(u,1)}$ on its support are precisely $C_u^{(1)}$ and $C_u^{(2)}$.

*Recovering the full tree.* Applying the preceding child-clade recovery argument at the root and then recursively to each non-singleton child clade recovers all clades of $\mathcal{T}$. A rooted tree with labeled leaves is uniquely determined by its clades via inclusion: $u'$ is a child of $u$ iff the clade of $u'$ is a maximal proper subset of the clade of $u$ (i.e., a strict subset $C' \subsetneq C$ with no clade $C''$ satisfying $C' \subsetneq C'' \subsetneq C$), and leaves are singleton clades. Thus $\mathcal{T}$ is recoverable from $V$. □

Here, we comment on child orderings. Different child orderings at node $u$ yield local bases that differ by an orthogonal transformation on $(S_u, \langle \cdot, \cdot \rangle_w)$. This induces an orthogonal transformation on the $(k_u - 1)$-dimensional coordinate block of $V$ corresponding to $u$. The geometry is preserved; only coordinate labels change. Now, we present the proof for Proposition 7.1.

**Proposition A.3** (Restatement of Proposition 7.1). *Let $\mathcal{L} = \mathbb{R}^d / \sim_\ell$ be the quotient space of logits under shift equivalence. Then $\mathcal{L} \cong \mathcal{H}$ (linear isomorphism). Concretely, for $\mathbf{p} = \mathrm{softmax}(\mathbf{z})$, we have*

$$\mathbf{z} - \bar{z}\mathbf{1} = \mathrm{clr}(\mathbf{p}), \quad \textit{where } \bar{z} = \tfrac{1}{d}\sum_i z_i,$$

*and $[\mathbf{z}] \rightarrow \mathbf{z} - \bar{z}\mathbf{1}$ is well-defined.*

*Proof.* We establish both the concrete identity and the isomorphism.

*(i) Centered logits equal CLR coordinates.* Let $p = \mathrm{softmax}(z)$, so $p_i = e^{z_i}/\sum_j e^{z_j}$. Taking logarithms,

$$\log p_i = z_i - \log \sum_j e^{z_j}.$$

The CLR transform is $\mathrm{clr}(p)_i = \log p_i - \frac{1}{d}\sum_k \log p_k$. Substituting,

$$\begin{aligned}
\mathrm{clr}(p)_i &= \left(z_i - \log \sum_j e^{z_j}\right) - \frac{1}{d}\sum_k \left(z_k - \log \sum_j e^{z_j}\right) \\
&= z_i - \log \sum_j e^{z_j} - \bar{z} + \log \sum_j e^{z_j} \\
&= z_i - \bar{z},
\end{aligned}$$

where $\bar{z} = \frac{1}{d}\sum_k z_k$. Thus $\mathrm{clr}(p) = z - \bar{z}\mathbf{1}$.

*(ii) Isomorphism.* Define the centering map $\pi\colon \mathbb{R}^d \to \mathcal{H}$ by $\pi(z) = z - \bar{z}\mathbf{1}$. We show $\pi$ induces a linear isomorphism from $\mathcal{L} = \mathbb{R}^d / \sim_\ell$ to $\mathcal{H}$. If $z' = z + c\mathbf{1}$ for some $c \in \mathbb{R}$, then $\bar{z}' = \bar{z} + c$, so

$$\pi(z') = z + c\mathbf{1} - (\bar{z} + c)\mathbf{1} = z - \bar{z}\mathbf{1} = \pi(z).$$

Thus $\pi$ is constant on equivalence classes and descends to a map $\tilde{\pi}\colon \mathcal{L} \to \mathcal{H}$. For any $z \in \mathbb{R}^d$, $\mathbf{1}^\top \pi(z) = \sum_i (z_i - \bar{z}) = d\bar{z} - d\bar{z} = 0$, so $\pi(z) \in \mathcal{H}$. Now, for any $h \in \mathcal{H}$, we have $\bar{h} = 0$, so $\pi(h) = h - 0 \cdot \mathbf{1} = h$. Thus $\pi$ is surjective onto $\mathcal{H}$. Suppose $\pi(z) = \pi(z')$. Then $z - \bar{z}\mathbf{1} = z' - \bar{z}'\mathbf{1}$, which gives $z - z' = (\bar{z} - \bar{z}')\mathbf{1}$. Thus $z \sim_\ell z'$, so $\tilde{\pi}$ is injective. Since $\pi$ is linear and constant on equivalence classes, the induced map $\tilde{\pi}$ is linear. Thus, $\tilde{\pi}\colon \mathcal{L} \to \mathcal{H}$ is a linear isomorphism. Moreover, equipping $\mathcal{L}$ with the quotient metric $d_\mathcal{L}([z], [z']) = \inf_{c \in \mathbb{R}} \|z - z' + c\mathbf{1}\|$, the infimum is attained at $c = \bar{z}' - \bar{z}$, which gives $d_\mathcal{L}([z], [z']) = \|\pi(z) - \pi(z')\|_2$. Thus $\tilde{\pi}$ is an isometry. □

### A.2. Algorithm Details

**Time complexity and runtime of PolyILR.** Algorithm 1 constructs $V$ in $O(d^2)$ time. For each internal node $u$ with $k_u$ children, building the Helmert matrix takes $O(k_u^2)$ time, Gram-Schmidt orthonormalization takes $O(k_u^3)$ time, and spreading to the $d$-dimensional vectors takes $O(n_u \cdot k_u)$ time where $n_u = \sum_r n_r$ is the number of leaves in the subtree rooted at $u$. Summing over all internal nodes, the total cost is dominated by the spreading step: $\sum_u n_u \cdot k_u \leq d \cdot \sum_u k_u = O(d^2)$ in the worst case (e.g., a star tree). For balanced trees, the complexity reduces to $O(d \log d)$. In practice, $V$ is computed once as a preprocessing step, so this cost is negligible compared to downstream tasks such as model training or statistical inference, which typically scale with sample size $N$ and involve iterative optimization. In all of our experiments, constructing $V$ using PolyILR took $< 5$ seconds.

**On Gram-Schmidt.**   The weighted Gram-Schmidt procedure orthonormalizes the columns of the Helmert matrix $H^{(u)}$ under the inner product $\langle h, h' \rangle_w = \sum_{r=1}^{k_u} h_r h'_r / n_r$. This is well-defined since $(S_u, \langle \cdot, \cdot \rangle_w)$ is a finite-dimensional Hilbert space. Explicitly, for $m = 1, \ldots, k_u - 1$:

$$\tilde{H}^{(u)}_{\cdot,m} = H^{(u)}_{\cdot,m} - \sum_{j=1}^{m-1} \langle H^{(u)}_{\cdot,m}, \tilde{H}^{(u)}_{\cdot,j} \rangle_w \, \tilde{H}^{(u)}_{\cdot,j}, \quad \text{then normalize: } \tilde{H}^{(u)}_{\cdot,m} \leftarrow \frac{\tilde{H}^{(u)}_{\cdot,m}}{\|\tilde{H}^{(u)}_{\cdot,m}\|_w}.$$

The weighting by $1/n_r$ accounts for clade sizes: larger clades contribute less per-component to the inner product, ensuring that the resulting ILR coordinates treat leaves uniformly regardless of tree imbalance.

**On Child ordering conventions.**   The child ordering at each internal node affects which contrasts appear in which columns of $V$, but does not affect the subspace spanned or the geometric properties as mentioned before. Two natural conventions are: (i) *Lexicographic:* Order children by the smallest leaf index in each subtree. This is deterministic given leaf labels. (ii) *By clade size:* Order children by descending $n_r$. This places contrasts involving larger clades in earlier columns. As noted in Proposition 4.2, different child orderings yield bases related by orthogonal transformations within each node's coordinate block. For reproducibility, we used the lexicographic ordering in all of our experiments.

**Reduction to PhILR.**   When the tree $\mathcal{T}$ is strictly binary (every internal node has exactly $k_u = 2$ children), PolyILR reduces to PhILR (Silverman et al., 2017). We verify this algebraically. For a binary node $u$ with child clades $C_u^{(1)}$ and $C_u^{(2)}$ of sizes $n_1$ and $n_2$, the Helmert matrix is $H^{(u)} = [1/\sqrt{2}, -1/\sqrt{2}]^\top$. Under the weighted inner product $\langle h, h' \rangle_w = h_1 h'_1 / n_1 + h_2 h'_2 / n_2$, the squared norm is

$$\|H^{(u)}\|^2_w = \frac{1/2}{n_1} + \frac{1/2}{n_2} = \frac{n_1 + n_2}{2 n_1 n_2}.$$

Applying Gram-Schmidt under $\langle \cdot, \cdot \rangle_w$ gives $\tilde{H}^{(u)} = [1/\sqrt{2}, -1/\sqrt{2}]^\top \cdot \sqrt{2 n_1 n_2 / (n_1 + n_2)}$. After spreading via $v_i = \tilde{H}_r / n_r$:

$$v_i^{(u)} = \begin{cases} +\sqrt{\dfrac{n_2}{n_1(n_1 + n_2)}} & \text{if } i \in C_u^{(1)}, \\[2ex] -\sqrt{\dfrac{n_1}{n_2(n_1 + n_2)}} & \text{if } i \in C_u^{(2)}, \\[2ex] 0 & \text{otherwise.} \end{cases}$$

This matches the PhILR balance formula exactly (Silverman et al., 2017). Thus PolyILR is a strict generalization of PhILR to arbitrary tree structures, with the binary case recovering the original method. We note that PhILR also supports optional weighting schemes: taxon weights (Egozcue & Pawlowsky-Glahn, 2016) that modify the simplex metric itself, and branch length weights that scale coordinates. These define a different but well-posed weighted geometry on the simplex (Egozcue & Pawlowsky-Glahn, 2016), rather than the standard Aitchison geometry. PolyILR is built on the standard (unweighted) Aitchison geometry to make the construction directly comparable to canonical ILR; integrating weighted-simplex variants is compatible in principle and left to future work.

# B. Additional Experiments and Details

## B.1. Microbiome Dataset and Taxonomy Construction Details

**Datasets.**   We use two microbiome datasets. HMP (Human Microbiome Project; 4,743 samples, 402 taxa) provides samples from 18 body sites with a curated NCBI-derived taxonomy. cMD3 (curatedMetagenomicData v3; 20,238 samples from 86 studies, 2,047 taxa) aggregates shotgun metagenomic data across diverse cohorts. Disease labels denote any non-healthy diagnosis, pooled across conditions and body sites. Note that 748 HMP samples also appear in cMD3; we do not deduplicate, as our focus is methodological comparison. The HMP data were obtained from the HMP16SData package in R. Operational taxonomic unit (OTU) abundances and sample metadata were extracted, and samples were filtered to those with complete body site annotations. The taxonomic tree was constructed from the NCBI-derived lineage strings provided with each OTU, parsed from Kingdom through Genus, and converted to Newick format using the ape package. The cMD3 metagenomic data were obtained from the curatedMetagenomicData package (v3.0) in R. We retrieved relative abundance data for all available studies, excluding IaniroG_2022. Taxonomic abundance matrices and sample

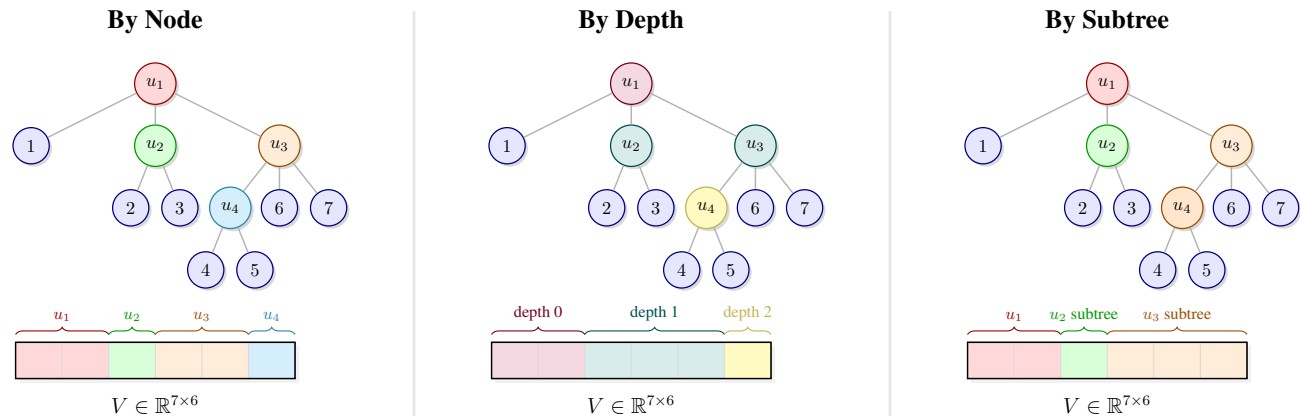

*Figure 5. Orthogonal subspace partitions.* The coordinates of $V$ can be grouped into orthogonal subspaces indexed by tree structure: by individual node, by depth level, or by subtree (one example). Each grouping enables inference at a different resolution.

metadata were extracted from `TreeSummarizedExperiment` objects and merged using `mergeData`. Only samples present in both the abundance matrix and metadata were retained. Taxonomic lineages were parsed from strings of the form `k__Kingdom|p__Phylum|c__Class|o__Order|f__Family|g__Genus|s__Species`, with rank prefixes removed. Taxon identifiers were generated via MD5 hashing (first 8 characters, prefixed `taxon_`) for reproducibility.

**Taxonomic Trees.** For `HMP`, we use the NCBI-derived taxonomy provided with the dataset. For `cMD3`, the taxonomic tree was constructed from lineage strings using the `data.tree` package with a common root, converted to Newick format via `ape`, and assigned ultrametric branch lengths using Grafen's method. Tip labels were replaced with hash-based identifiers to match the abundance matrix. Because `cMD3` aggregates studies with heterogeneous taxonomic resolution, some internal nodes have children spanning multiple taxonomic ranks; we label such nodes by their lowest common rank (e.g., *Bacteria (Kingdom)* or *Mixed*).

### B.2. Single-Cell Dataset and Ontology Construction Details

**Single-Cell Dataset.** We use `DISCO` (Database of Immune Single-Cell Omics), a curated atlas of human immune single-cell transcriptomes. Data were extracted via the `DISCOtoolkit` R package. We selected samples with $\geq 500$ cells and randomly sampled up to 200 samples per condition. We extracted blood tissue (healthy, COVID-19, leukemia) and liver tissue (healthy, hepatocellular carcinoma). Cell type proportions were computed by counting cells per annotated type and normalizing to sum to one, following standard practice in single-cell compositional analysis (Phipson et al., 2022; Buettner et al., 2021). After processing: blood (199 healthy, 200 COVID-19, 199 leukemia; 62 cell types) and liver (200 healthy, 153 HCC; 99 cell types). For the main experiments, we use the leukemia and HCC tasks.

**Cell Ontology Tree.** The cell type hierarchy was obtained from DISCO's cell ontology API (Li et al., 2022). For each tissue, we constructed a subtree by: (1) identifying cell types present in the data, (2) filtering to true leaves (types with no children in the data), and (3) tracing ancestor paths to the root. Unit branch lengths were assigned. The cell ontology contains extensive polytomies: blood has 14 nodes with >2 children ("T cell" has 9 children); liver has similar structure with 99 leaves and 72 internal nodes.

### B.3. Software and Hyperparameters

Classification models were tuned via 5-fold cross-validation on training data. Grid search ranges and selected values are summarized in Table 7. Feature importance was computed via mean decrease in impurity (random forest), coefficient magnitude (logistic regression), or permutation importance (SVM).

We compare PolyILR against: (i) **CLR** (centered log-ratio), which ignores tree structure; and (ii) **PhILR** (Silverman et al., 2017), which requires arbitrary binary refinement of polytomies. Both use identical zero handling.

*Table 7.* Hyperparameter search grid and selected values.

| Model | Parameter | Search Range | Used |
|---|---|---|---|
| Random Forest | `n_estimators` | $\{100, 300, 500\}$ | 500 |
| | `max_depth` | $\{10, 20, \texttt{None}\}$ | 20 |
| | `max_features` | $\{\texttt{sqrt}, \texttt{log2}\}$ | `sqrt` |
| Logistic Reg. | $C$ (inverse reg.) | $\{0.01, 0.1, 1, 10\}$ | 1 |
| | penalty | $\{\ell_1, \ell_2\}$ | $\ell_2$ |
| SVM (RBF) | $C$ | $\{0.1, 1, 10\}$ | 1 |
| | $\gamma$ | $\{\texttt{scale}, 0.01, 0.1\}$ | `scale` |

## B.4. Handling Zeros in PolyILR

The raw compositional data (e.g., microbiome counts, single-cell type counts) typically exhibit high sparsity, with zeros comprising 70% or more of entries. Since log-ratio transformations are undefined at zero, zero replacement is required as preprocessing. There is no consensus on the optimal choice; common choices include 0.5, 1, or smaller values, depending on whether the raw data are counts or proportions (Kaul et al., 2017; Hu et al., 2022). Because our focus is on comparing representations rather than zero-handling strategies, we adopt simple per-dataset choices appropriate to each data scale as follows. `HMP` and `cMD3` are count data: `HMP` provides raw 16S counts, and `cMD3` (curatedMetagenomicData v3) is retrieved with `counts=TRUE`, which multiplies relative abundances by read depth. For both, we add a pseudocount of 1 prior to normalization, i.e., $\tilde{x}_i = (x_i + 1)/\sum_j (x_j + 1)$, which is commonly done at count scale. For `cMD3`, one study (`IaniroG_2022`) was excluded because read depth metadata was unavailable. *DISCO* provides cell-type proportions rather than counts, so we instead use $\epsilon = 10^{-10}$ applied additively before renormalization. We note that PolyILR partially mitigates the effect of zero replacement, because each coordinate is a contrast between *geometric means over clades* rather than between individual taxa (Section 4.2), but a rigorous study is for future work.

## B.5. PolyILR's Robustness under Topological Noise

As discussed in Section 9, PolyILR assumes a fixed input tree $\mathcal{T}$, and its coordinates are defined relative to $\mathcal{T}$. This is reasonable because PolyILR is designed for *curated scientific hierarchies* provided by domain experts (e.g., phylogenies, taxonomies, or ontologies) which can largely be trusted. In practice, however, some tree noise or error is unavoidable. One common form is *topological noise*, e.g., small misestimations of branching order or leaf placement. Therefore, we provide a preliminary analysis of how the PolyILR basis and resulting representations respond to such local perturbations in the input tree, both empirically (as semantic stability of selected features) and structurally (as how much of the basis $V$ is preserved).

We introduce topological noise via random *nearest-neighbor interchange* (NNI) operations on the NCBI taxonomy of the `HMP` dataset. Each NNI swaps two subtrees across an internal edge, locally rearranging the tree without changing leaf labels. We report perturbation strength as the absolute number of NNI applied randomly. We use the `HMP` body-subsite task (18 classes, $d = 402$ taxa), matching the setting of Table 2 (bottom).

| #NNIs | PolyILR semantic top-10 stability |
|---|---|
| 0 | $0.73 \pm 0.10$ |
| 1 | $0.71 \pm 0.11$ |
| 2 | $0.68 \pm 0.09$ |
| 3 | $0.65 \pm 0.08$ |

*Table 8.* Semantic Jaccard similarity of top-10 PolyILR features under NNI perturbations on the `HMP` taxonomy (18 body subsites, 402 taxa). Mean $\pm$ std over 10 perturbed trees $\times$ 10 seeds.

**Semantic stability under NNI.** For each NNI count, we perturb the original tree, recompute the PolyILR basis $V$, retrain the downstream RF model, and re-extract the top-10 important features. We then measure semantic Jaccard similarity of the top-10 features against the unperturbed baseline, averaged over 10 perturbed trees and 10 random seeds (Table 8). Other setups mirror those used for Table 2. At 0 NNIs (no noise), the result matches Table 2 (bottom). As we introduce more NNIs, PolyILR degrades smoothly but *remains substantially more stable than PhILR* even at 3 NNIs: PolyILR at 3 NNIs (0.65) is still well above PhILR at 0 NNIs (0.13, from Table 2 bottom), which suffers from binarization-induced instability on top of any tree noise. This shows that PolyILR remains effective even at moderate noise levels.

**Locality of NNI effects on the basis $V$.** The robustness above has a structural explanation: the PolyILR basis factors node by node. For each internal node $u$ with children $c_1, \ldots, c_{k_u}$, $V$ contains exactly $k_u - 1$ local basis vectors associated with $u$. Once child ordering is fixed, this local block depends only on the partition of descendant leaves induced by the children of $u$ and their subtree sizes $\{n_r\}$ (Section 4.2). After spreading, each basis is supported only on the leaves descending from $u$, and no node's construction references any other node's local structure. Therefore, a local topological change has only a local effect on $V$.

Consider an NNI at an internal edge $(p, c)$, where $p$ is the parent of $c$. At $p$ and $c$, the child partitions directly change, so the local coordinate blocks of $p$ and $c$ in $V$ also change; above $p$, the swap happens entirely within $p$'s subtree, so the descendant leaf sets under each ancestor's children, and their subtree sizes, are unchanged, so their local blocks in $V$ are *exactly preserved*; below $c$, the internal structure of each subtree is untouched, so again their local blocks in $V$ are *exactly preserved*. This way, an NNI perturbation modifies only the coordinate blocks attached to a small set of affected nodes; all others remain preserved.

| #NNIs | Cols at perturbed nodes | Cols changed | Cols preserved (%) |
|---|---|---|---|
| 1 | 18.8 | 8.4 | 392.6 / 401 (97.9%) |
| 2 | 26.6 | 13.2 | 387.8 / 401 (96.7%) |
| 3 | 36.6 | 18.6 | 382.4 / 401 (95.4%) |

*Table 9.* NNI effects on $V$ (HMP, 402 taxa, 401 coords; mean over 5 seeds). We report the number of cumulative NNIs, the numbers of columns at perturbed nodes, of columns that changed, and of columns that are preserved.

We verify this empirically on the HMP taxonomy (402 taxa, 401 PolyILR coordinates) above. For each NNI count, we apply $\{1, 2, 3\}$ cumulative random NNI moves and compare $V$ before and after, averaged over 5 seeds. In Table 9, we observe that the number of changed columns never exceeds the number of columns at perturbed nodes, and most of $V$ remains exactly preserved. This empirically supports the structural argument and helps explain the mild degradation seen in Table 8.

## B.6. Extended Classification Results

We provide a preliminary empirical illustration of the geometric connection described in Section 7. Using CIFAR-100 with its known 20-superclass hierarchy and a ResNet-34 trained to 80.4% test accuracy, we ask two questions: (i) Does the true semantic hierarchy capture structure in the model's error distribution that arbitrary hierarchies do not? (ii) If so, how does this structure develop during training?

**Setup and discussion.** The CIFAR-100 hierarchy has 100 fine classes grouped into 20 superclasses, yielding a two-level tree with 19 depth-0 coordinates (superclass contrasts) and 80 depth-1 coordinates (within-superclass contrasts). For each test sample, we compute the gradient $\nabla_{\mathbf{a}} \ell = \mathbf{V}^\top (\mathbf{p} - \mathbf{e}_y)$ in PolyILR coordinates and measure how gradient norm distributes across depths. We summarize this via *gradient entropy*: $H = -\sum_d p_d \log p_d$, where $p_d$ is the fraction of total gradient norm at depth $d$. Lower entropy indicates concentration at specific depths, which we view as *some structure* emerging.

*(i) Known tree vs. shuffled trees.* We compare the true hierarchy against 1000 random trees constructed by permuting leaf assignments while preserving structure. For the trained model, the true tree yields significantly lower entropy (0.576 vs. $0.632 \pm 0.001$; $p < 0.001$), indicating that errors concentrate at specific depths under the true hierarchy but not under arbitrary ones. For a randomly initialized model, no difference exists ($p = 0.73$). This confirms that this structure emerges from learning rather than architectural bias. This aligns with recent work showing that flat classifiers implicitly encode semantic hierarchies recoverable from logits alone (Palumbo et al., 2025).

*(ii): Learning dynamics.* We track gradient distribution across 200 training epochs. The fraction at depth 1 (within-superclass) increases from 0.68 to 0.74 over training, consistent with the model progressively resolving coarse superclass distinctions before fine-grained class boundaries. This coarse-to-fine pattern—recently termed *hypernym bias* (Malashin et al., 2025)—reflects curriculum-like learning where easier (coarser) distinctions are learned first.

**Implication.** These results suggest that PolyILR coordinates may provide a meaningful lens for analyzing softmax classifiers and steering model training when class hierarchies are available. The gradient decomposition partly reveals where in the hierarchy a model's errors concentrate and how this evolves during training. We view this as opening a research direction rather than a complete empirical study, which we leave to future work.

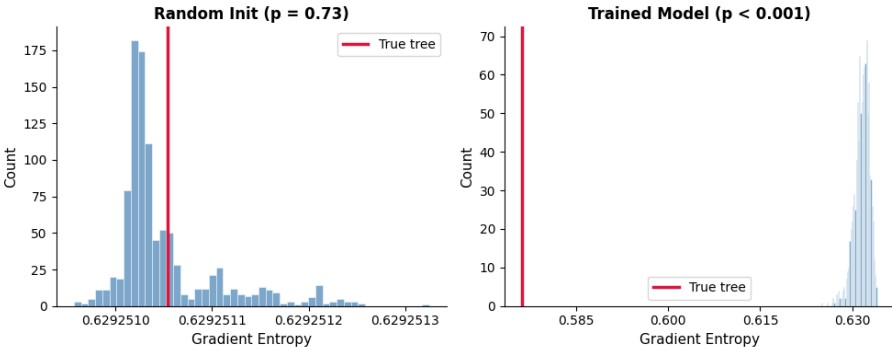

*Figure 6.* Learning dynamics on `CIFAR-100`. **Left:** Gradient norm distribution by tree depth across training. Depth 0 (superclass contrasts) decreases while depth 1 (within-superclass) increases, indicating coarse-to-fine learning. **Right:** Gradient entropy and test accuracy over training epochs.

### B.7. Full Experimental Results

This appendix provides complete experimental results; biological interpretation is in Section 6. The patterns are consistent with those discussed in the main text: PolyILR recovers interpretable taxonomic and ontological contrasts across all tasks. Deeper biological validation—including wet-lab experiments and large-scale cohort studies—is beyond the scope of this methodological work. Throughout these tables, we report point estimates alongside variability (e.g., 95% confidence intervals for accuracy and AUROC, std for importance, and rank range across CV folds), which are small in our observations and demonstrate robustness of the reported numbers.

Table 10 reports classification accuracy and AUROC with 95% CIs across all tasks: body sites/subsites (`HMP`), westernization/age/health/body site (`cMD3`), and COVID-19(blood)/leukemia(blood)/HCC(liver) (`DISCO`). SVM and LR are identical across CLR, PhILR, and PolyILR within each task, as expected from isometry; RF varies modestly. Table 11 reports feature stability (Jaccard similarity of top-$K$ features across CV folds); PolyILR remains stable while PhILR suffers from arbitrary binarization for both index and semantic stabilities. Tables 12–14 list the top-10 PolyILR contrasts by RF importance for `HMP` (microbiome, body sites), `cMD3` (microbiome, health/lifestyle), and `DISCO` (single-cell, disease), with mean importance, std, and rank range across 5-fold CV. Each contrast is a log-ratio of geometric means between two groups at a tree node. Tables 15–17 report tree-level aggregation results, with importance aggregated by depth (cumulative), subtree, node, and taxon/cell-type levels. Accuracy columns show predictive performance using only features at that level.

*Table 10. Classification accuracy and AUROC with 95% confidence intervals* (all tasks). Each cell reports RF / SVM / LR. SVM and LR are identical across CLR, PhILR, and PolyILR within each task, as expected from isometry; RF varies modestly. Upper CI bounds clipped at 1.00.

| Task | | CLR | PhILR | PolyILR |
|---|---|---|---|---|
| | | RF / SVM / LR | RF / SVM / LR | RF / SVM / LR |
| *Acc (%) (95% CI)* | | | | |
| HMP | body sites (5) | .956 (.952–.959) / .971 (.968–.974) / .962 (.956–.968) | .961 (.958–.965) / .971 (.968–.974) / .962 (.955–.968) | .963 (.961–.966) / .971 (.968–.974) / .962 (.955–.968) |
| | body subsites (18) | .597 (.584–.610) / .672 (.660–.685) / .646 (.637–.655) | .608 (.588–.627) / .672 (.660–.685) / .646 (.638–.654) | .622 (.608–.635) / .672 (.660–.685) / .646 (.637–.656) |
| cMD3 | westernized (2) | .972 (.967–.977) / .979 (.972–.986) / .968 (.956–.979) | .966 (.959–.972) / .979 (.972–.986) / .967 (.955–.979) | .967 (.962–.972) / .979 (.972–.986) / .967 (.956–.979) |
| | age category (5) | .785 (.755–.815) / .814 (.796–.833) / .739 (.712–.766) | .797 (.782–.813) / .814 (.796–.833) / .738 (.712–.765) | .797 (.784–.811) / .814 (.796–.833) / .738 (.711–.765) |
| | healthy/disease (2) | .692 (.665–.719) / .701 (.681–.721) / .664 (.638–.690) | .686 (.651–.721) / .702 (.682–.721) / .664 (.638–.690) | .681 (.648–.714) / .702 (.682–.721) / .664 (.638–.690) |
| | disease stool (2) | .688 (.668–.707) / .691 (.662–.721) / .662 (.623–.701) | .677 (.643–.710) / .691 (.662–.721) / .663 (.624–.701) | .674 (.640–.707) / .691 (.662–.721) / .663 (.624–.701) |
| | body site (6) | .982 (.976–.988) / .989 (.984–.994) / .987 (.983–.991) | .987 (.980–.993) / .989 (.984–.994) / .987 (.983–.991) | .987 (.982–.992) / .989 (.984–.994) / .987 (.983–.991) |
| DISCO | COVID-19 (2) | .747 (.661–.832) / .777 (.688–.866) / .737 (.679–.794) | .762 (.668–.856) / .777 (.688–.866) / .737 (.679–.794) | .777 (.688–.865) / .777 (.688–.866) / .739 (.683–.796) |
| | leukemia (2) | .925 (.883–.967) / .932 (.893–.971) / .927 (.880–.974) | .940 (.900–.980) / .932 (.893–.971) / .927 (.880–.974) | .935 (.891–.978) / .932 (.893–.971) / .927 (.880–.974) |
| | HCC (2) | .921 (.807–1.00) / .927 (.835–1.00) / .944 (.863–1.00) | .910 (.794–1.00) / .927 (.835–1.00) / .944 (.863–1.00) | .910 (.800–1.00) / .927 (.835–1.00) / .944 (.863–1.00) |
| *AUROC (95% CI)* | | | | |
| HMP | body sites (5) | .987 (.981–.992) / .995 (.994–.996) / .994 (.993–.996) | .992 (.991–.994) / .995 (.994–.996) / .994 (.993–.996) | .992 (.990–.994) / .995 (.994–.996) / .994 (.993–.996) |
| | body subsites (18) | .957 (.954–.960) / .974 (.973–.975) / .967 (.966–.969) | .965 (.962–.967) / .974 (.973–.975) / .967 (.966–.969) | .966 (.964–.969) / .974 (.973–.975) / .967 (.966–.969) |
| cMD3 | westernized (2) | .975 (.965–.986) / .978 (.958–.999) / .967 (.940–.995) | .966 (.951–.980) / .978 (.958–.999) / .966 (.938–.994) | .966 (.951–.982) / .978 (.958–.999) / .967 (.939–.995) |
| | age category (5) | .836 (.802–.870) / .867 (.838–.895) / .833 (.811–.855) | .837 (.804–.871) / .867 (.838–.895) / .833 (.811–.855) | .843 (.813–.873) / .867 (.838–.895) / .833 (.811–.855) |
| | healthy/disease (2) | .662 (.634–.691) / .690 (.665–.715) / .629 (.598–.660) | .637 (.607–.668) / .690 (.665–.715) / .629 (.598–.660) | .631 (.596–.667) / .690 (.665–.715) / .629 (.598–.660) |
| | disease stool (2) | .663 (.630–.696) / .683 (.645–.722) / .633 (.587–.680) | .634 (.590–.678) / .683 (.645–.722) / .633 (.587–.680) | .632 (.586–.677) / .683 (.645–.722) / .633 (.587–.680) |
| | body site (6) | .945 (.907–.982) / .994 (.991–.997) / .996 (.993–.999) | .998 (.996–1.00) / .994 (.992–.997) / .996 (.993–.999) | .981 (.949–1.00) / .994 (.992–.997) / .996 (.993–.999) |
| DISCO | COVID-19 (2) | .808 (.699–.918) / .897 (.818–.977) / .847 (.784–.910) | .875 (.802–.948) / .897 (.818–.977) / .847 (.784–.911) | .882 (.806–.957) / .897 (.817–.977) / .848 (.785–.911) |
| | leukemia (2) | .968 (.946–.990) / .984 (.969–.999) / .982 (.963–1.00) | .984 (.964–1.00) / .984 (.969–.999) / .982 (.963–1.00) | .984 (.966–1.00) / .984 (.969–.999) / .982 (.963–1.00) |
| | HCC (2) | .921 (.794–1.00) / .996 (.990–1.00) / .998 (.996–1.00) | .984 (.960–1.00) / .996 (.990–1.00) / .998 (.996–1.00) | .974 (.935–1.00) / .996 (.990–1.00) / .998 (.996–1.00) |

*Table 11. Feature stability*: Jaccard similarity of top-$K$ features across CV folds (all tasks). PolyILR stable; PhILR (index/semantic) unstable from arbitrary binarization.

| | Task | PolyILR | | | PhILR (index / semantic) | | |
|---|---|---|---|---|---|---|---|
| | | $K=5$ | $K=10$ | $K=50$ | $K=5$ | $K=10$ | $K=50$ |
| HMP | body sites (5) | .66 | .65 | .84 | .01 / .08 | .01 / .06 | .07 / .05 |
| | body subsites (18) | .71 | .72 | .88 | .01 / .22 | .02 / .13 | .07 / .04 |
| cMD3 | westernized (2) | .43 | .69 | .81 | .00 / .01 | .00 / .01 | .01 / .02 |
| | age category (5) | .73 | .58 | .76 | .12 / .04 | .09 / .03 | .03 / .03 |
| | healthy/disease (2) | .56 | .86 | .80 | .00 / .02 | .00 / .03 | .02 / .02 |
| | disease stool (2) | .53 | .67 | .80 | .00 / .02 | .00 / .03 | .02 / .03 |
| | body site (6) | .33 | .32 | .62 | .00 / .00 | .00 / .00 | .01 / .01 |
| DISCO | COVID-19 (2) | 1.0 | .87 | .99 | .06 / .07 | .12 / .11 | .73 / .12 |
| | leukemia (2) | .75 | .81 | .92 | .05 / .09 | .12 / .16 | .73 / .12 |
| | HCC (2) | .77 | .78 | .84 | .03 / .07 | .06 / .09 | .35 / .11 |

*Table 12. Top-10 contrasts* by RF importance (%) with mean $\pm$ std and rank range across 5-fold CV: HMP.

| | Rk | Contrast | Imp. (mean $\pm$ std) | Rank range |
|---|---|---|---|---|
| body sites (5 classes) | 1 | Streptococcaceae vs Lactobacillus + Leuconostocaceae | 3.94 ± 0.06 | 1 |
| | 2 | Lactococcus vs Streptococcus | 3.42 ± 0.16 | 2–3 |
| | 3 | Pseudomonadales vs Cardiobacteriaceae + Vibrionaceae + Legionellales + ... | 3.40 ± 0.17 | 2–3 |
| | 4 | Bacillales vs Gemella + Exiguobacterium + Turicibacter + ... | 2.95 ± 0.07 | 4–5 |
| | 5 | Pasteurella vs Haemophilus + Actinobacillus + Aggregatibacter | 2.85 ± 0.08 | 4–5 |
| | 6 | Veillonellaceae vs Dehalobacterium + Fusibacter + Peptococcus + ... | 2.61 ± 0.06 | 6 |
| | 7 | Pasteurellaceae vs Cardiobacteriaceae + Vibrionaceae + Legionellales + ... | 2.37 ± 0.06 | 7–9 |
| | 8 | Leuconostocaceae vs Lactobacillus | 2.32 ± 0.14 | 7–9 |
| | 9 | Actinomycetaceae vs Streptomyces + Microbispora + Brevibacterium + ... | 2.29 ± 0.08 | 7–9 |
| | 10 | Propionibacteriaceae vs Streptomyces + Microbispora + Brevibacterium + ... | 1.81 ± 0.17 | 10–17 |
| body subsites (18 classes) | 1 | Oribacterium vs Blautia + Coprococcus + Eubacterium + ... | 1.22 ± 0.03 | 1 |
| | 2 | Clostridiales vs Bacilli | 1.10 ± 0.07 | 2–4 |
| | 3 | Streptococcaceae vs Lactobacillus + Leuconostocaceae | 1.08 ± 0.04 | 2–5 |
| | 4 | Lactococcus vs Streptococcus | 1.03 ± 0.03 | 3–5 |
| | 5 | Corynebacterium vs Streptomyces + Microbispora + Brevibacterium + ... | 1.02 ± 0.03 | 3–5 |
| | 6 | Bacillales vs Gemella + Exiguobacterium + Turicibacter + ... | 0.92 ± 0.02 | 6–8 |
| | 7 | Propionibacteriaceae vs Streptomyces + Microbispora + Brevibacterium + ... | 0.91 ± 0.01 | 6–8 |
| | 8 | Pseudonocardiaceae vs Streptomyces + Microbispora + Brevibacterium + ... | 0.89 ± 0.05 | 6–12 |
| | 9 | Ruminococcaceae vs Dehalobacterium + Fusibacter + Peptococcus + ... | 0.88 ± 0.01 | 8–11 |
| | 10 | Propionibacterium vs Tessaracoccus | 0.85 ± 0.04 | 7–12 |

*Table 13. Top-10 contrasts* by RF importance (%) with mean ± std and rank range across 5-fold CV: `cMD3`.

| | Rk | Contrast | Imp. (mean ± std) | Rank range |
|---|---|---|---|---|
| westernized (2 classes) | 1 | Prevotella vs Alistipes + Anaerotruncus + Bacteroides + ... | 1.77 ± 0.12 | 1 |
| | 2 | Murimonas vs Alistipes + Anaerotruncus + Bacteroides + ... | 1.49 ± 0.07 | 2–4 |
| | 3 | Prevotella vs Bacteroides + Alistipes | 1.45 ± 0.03 | 2–5 |
| | 4 | Lactobacillus vs Alistipes + Anaerotruncus + Bacteroides + ... | 1.38 ± 0.04 | 4–6 |
| | 5 | Bacteroides vs Alistipes + Anaerotruncus + Bacteroides + ... | 1.36 ± 0.11 | 3–8 |
| | 6 | Dialister vs Alistipes + Anaerotruncus + Bacteroides + ... | 1.32 ± 0.15 | 2–13 |
| | 7 | Tissierellia_unclassified vs Alistipes + Anaerotruncus + Bacteroides + ... | 1.31 ± 0.09 | 5–10 |
| | 8 | Peptoniphilus vs Alistipes + Anaerotruncus + Bacteroides + ... | 1.28 ± 0.05 | 5–10 |
| | 9 | Megasphaera vs Alistipes + Anaerotruncus + Bacteroides + ... | 1.23 ± 0.04 | 6–10 |
| | 10 | Prevotella vs Citrobacter + Bacteroides + Leuconostoc + ... | 1.18 ± 0.06 | 7–12 |
| age category (5 classes) | 1 | Mixed vs Phascolarctobacterium + Tychonema + Sediminibacterium + ... | 1.32 ± 0.04 | 1–2 |
| | 2 | Bacteria vs Bacteria + Bacteria + Bacteria + ... | 1.24 ± 0.07 | 1–3 |
| | 3 | Mixed vs Bacteria + Mixed | 1.21 ± 0.07 | 2–3 |
| | 4 | Mixed vs Phascolarctobacterium + Tychonema + Sediminibacterium + ... | 0.90 ± 0.03 | 4 |
| | 5 | Holdemanella vs Faecalibacterium | 0.75 ± 0.04 | 5–10 |
| | 6 | Enterococcus vs Blautia | 0.73 ± 0.06 | 5–9 |
| | 7 | Mixed vs Dialister + Bacteria + Mixed + ... | 0.69 ± 0.03 | 6–8 |
| | 8 | Mixed vs Mixed | 0.69 ± 0.03 | 5–9 |
| | 9 | Bacteria vs Dialister + Bacteria + Mixed + ... | 0.67 ± 0.05 | 7–14 |
| | 10 | Mixed vs Bacteria + Bacteria + Bacteria + ... | 0.66 ± 0.04 | 6–11 |
| healthy vs disease (2) | 1 | Lachnoclostridium vs Bacteroides + Turicibacter + Ruminococcus + ... | 0.55 ± 0.06 | 1 |
| | 2 | Bifidobacterium vs Blautia + Enterococcus | 0.38 ± 0.04 | 2–5 |
| | 3 | Flavonifractor vs Bifidobacterium + Actinomyces | 0.35 ± 0.02 | 2–6 |
| | 4 | Fusicatenibacter vs Gemmiger | 0.34 ± 0.02 | 2–6 |
| | 5 | Enterococcus vs Blautia | 0.34 ± 0.02 | 3–6 |
| | 6 | Lachnoclostridium vs Parabacteroides + Bacteroides + Barnesiella + ... | 0.33 ± 0.02 | 4–8 |
| | 7 | Coprococcus vs Ruthenibacterium + Coprobacter + Oscillibacter + ... | 0.30 ± 0.02 | 3–9 |
| | 8 | Actinomyces vs Bifidobacterium + Actinomyces + Flavonifractor | 0.29 ± 0.02 | 6–10 |
| | 9 | Streptococcus vs Enterococcus + Stenotrophomonas + Clostridioides | 0.28 ± 0.02 | 8–10 |
| | 10 | Gemella vs Aeriscardovia + Enterococcus + Dickeya + ... | 0.27 ± 0.02 | 7–16 |
| stool disease (2 classes) | 1 | Lachnoclostridium vs Bacteroides + Turicibacter + Ruminococcus + ... | 0.58 ± 0.04 | 1 |
| | 2 | Gemella vs Aeriscardovia + Enterococcus + Dickeya + ... | 0.38 ± 0.03 | 2–5 |
| | 3 | Bifidobacterium vs Blautia + Enterococcus | 0.37 ± 0.02 | 2–4 |
| | 4 | Fusicatenibacter vs Gemmiger | 0.33 ± 0.02 | 4–10 |
| | 5 | Lachnoclostridium vs Parabacteroides + Bacteroides + Barnesiella + ... | 0.33 ± 0.03 | 3–11 |
| | 6 | Flavonifractor vs Bifidobacterium + Actinomyces | 0.32 ± 0.02 | 5–8 |
| | 7 | Enterococcus vs Blautia | 0.31 ± 0.02 | 5–8 |
| | 8 | Actinomyces vs Bifidobacterium + Actinomyces + Flavonifractor | 0.31 ± 0.04 | 4–14 |
| | 9 | Streptococcus vs Enterococcus + Stenotrophomonas + Clostridioides | 0.30 ± 0.03 | 5–11 |
| | 10 | Coprococcus vs Ruthenibacterium + Coprobacter + Oscillibacter + ... | 0.29 ± 0.03 | 3–14 |
| body site (6 classes) | 1 | Bacillus vs Aggregatibacter + Prevotella + Leptotrichia + ... | 1.43 ± 0.05 | 1–2 |
| | 2 | Bacteria vs Dialister + Bacteria + Mixed + ... | 1.30 ± 0.12 | 1–4 |
| | 3 | Cutibacterium vs Parascardovia + Streptococcus + Erysipelatoclostridium + ... | 1.30 ± 0.05 | 2–3 |
| | 4 | Arcobacter vs Aggregatibacter + Prevotella + Leptotrichia + ... | 1.21 ± 0.14 | 2–7 |
| | 5 | Lactobacillus vs Aggregatibacter + Prevotella + Leptotrichia + ... | 1.05 ± 0.05 | 6–11 |
| | 6 | Mixed vs Phascolarctobacterium + Tychonema + Sediminibacterium + ... | 1.04 ± 0.10 | 5–15 |
| | 7 | Bacteria vs Bacteria | 1.04 ± 0.03 | 4–11 |
| | 8 | Cand. Gastranaerophilales vs Aggregatibacter + Prevotella + Leptotrichia + ... | 1.02 ± 0.06 | 6–13 |
| | 9 | Providencia vs Aggregatibacter + Prevotella + Leptotrichia + ... | 1.00 ± 0.08 | 8–19 |
| | 10 | Helicobacter vs Aggregatibacter + Prevotella + Leptotrichia + ... | 0.99 ± 0.04 | 5–14 |

*Table 14. Top-10 contrasts* by RF importance (%) with mean ± std and rank range across 5-fold CV: `DISCO`.

| | Rk | Contrast | Imp. (mean ± std) | Rank range |
|---|---|---|---|---|
| COVID-19 (blood) | 1 | Treg cell vs Naive CD4 T cell + Memory CD4 T cell + Tfh cell | 9.03 ± 0.88 | 1 |
| | 2 | CD4 T cell vs Naive T cell + Memory T cell + CD8 T cell + ... | 7.34 ± 0.64 | 2–4 |
| | 3 | Cycling T/NK cell vs ILC + T cell + NK cell | 6.68 ± 0.43 | 2–4 |
| | 4 | Tfh cell vs Naive CD4 T cell + Memory CD4 T cell | 6.46 ± 0.28 | 3–4 |
| | 5 | Plasma cell vs Pre-GC B cell + B cell precursor + Memory B cell + ... | 5.03 ± 0.26 | 5 |
| | 6 | INF-activated T cell vs Naive T cell + Memory T cell + CD8 T cell + ... | 3.96 ± 0.35 | 6–8 |
| | 7 | Naive CD8 T cell vs Memory CD8 T cell | 3.73 ± 0.48 | 6–8 |
| | 8 | Double negative T cell vs Naive T cell + Memory T cell + CD8 T cell + ... | 2.93 ± 0.29 | 8–9 |
| | 9 | Gamma delta T cell vs Naive T cell + Memory T cell + CD8 T cell + ... | 2.90 ± 0.73 | 6–18 |
| | 10 | Epithelial cell vs Immune cell + Neuron | 2.37 ± 0.18 | 9–12 |
| leukemia (blood) | 1 | Myeloid cell vs Erythrocyte/Megakaryocyte + Hematopoietic precursor cell | 14.50 ± 0.46 | 1 |
| | 2 | Cycling T/NK cell vs ILC + T cell + NK cell | 11.79 ± 0.54 | 2 |
| | 3 | Lymphoid cell vs Erythrocyte/Megakaryocyte + Hematopoietic precursor + Myeloid | 6.30 ± 0.45 | 3–4 |
| | 4 | MAIT cell vs Naive T cell + Memory T cell + CD8 T cell + ... | 5.45 ± 0.51 | 3–5 |
| | 5 | Naive CD8 T cell vs Memory CD8 T cell | 4.73 ± 0.87 | 3–8 |
| | 6 | Monocyte vs Granulocyte | 4.22 ± 0.37 | 6–7 |
| | 7 | T/NK cell vs B cell | 4.09 ± 0.67 | 5–8 |
| | 8 | Dendritic cell vs Granulocyte + Monocyte | 3.13 ± 0.44 | 7–10 |
| | 9 | Gamma delta T cell vs Naive T cell + Memory T cell + CD8 T cell + ... | 2.83 ± 0.33 | 7–13 |
| | 10 | Cycling T cell vs Naive T cell + Memory T cell + CD8 T cell | 2.66 ± 0.42 | 8–14 |
| HCC (liver) | 1 | Erythrocyte/Megakaryocyte vs Myeloid + Lymphoid + Hematopoietic precursor | 10.08 ± 0.32 | 1 |
| | 2 | B cell precursor vs Plasma cell + INF-activated naive B cell + Memory B cell + ... | 6.52 ± 0.32 | 2–4 |
| | 3 | Hematopoietic precursor cell vs Myeloid cell + Lymphoid cell | 6.34 ± 0.55 | 2–3 |
| | 4 | Venous EC vs LSEC | 5.45 ± 0.85 | 2–8 |
| | 5 | Granulocyte-monocyte progenitor vs Multipotent progenitor + Megakaryocyte + ... | 4.37 ± 0.51 | 5–7 |
| | 6 | Immune cell vs Cycling villous cytotrophoblast + CCL19/21 pericyte + Muscle + ... | 4.37 ± 0.39 | 4–8 |
| | 7 | MHCII high CD14 monocyte vs MHCII low CD14 monocyte | 3.97 ± 0.38 | 5–10 |
| | 8 | Intermediate EPCAM+ erythroblast vs Late hemoglobin+ erythroblast | 3.78 ± 0.40 | 6–10 |
| | 9 | Alveolar macrophage vs Kupffer cell | 3.52 ± 0.27 | 7–10 |
| | 10 | CD14 monocyte vs Placenta defensin+ monocyte + Placenta CAMP+ monocyte + ... | 3.22 ± 0.34 | 8–12 |

*Table 15. Tree-level inference*: `HMP`. Importance (mean ± std) aggregated by depth (cumulative), subtree (phylum), node, and taxon. Accuracy uses only features at that level.

| Structure | | body sites (5) | | | body subsites (18) | | |
|---|---|---|---|---|---|---|---|
| | Component | Acc | Imp | Component | Acc | Imp |
| Depth | ≤0 (coarsest) | .87 ± .01 | 6.8 ± 0.2% | ≤0 (coarsest) | .42 ± .01 | 4.8 ± 0.1% |
| | ≤1 | .92 ± .01 | 12 ± 0.2% | ≤1 | .52 ± .01 | 12 ± 0.2% |
| | ≤2 | .96 ± .01 | 47 ± 0.3% | ≤2 | .59 ± .03 | 44 ± 0.3% |
| | ≤3 | .96 ± .01 | 86 ± 0.5% | ≤3 | .62 ± .02 | 87 ± 0.0% |
| | ≤4 (all) | .96 ± .00 | 100% | ≤4 (all) | .62 ± .01 | 100% |
| Subtree | Firmicutes | .95 ± .01 | 47 ± 0.7% | Firmicutes | .55 ± .02 | 36 ± 0.3% |
| | Proteobacteria | .87 ± .01 | 20 ± 0.3% | Proteobacteria | .44 ± .02 | 27 ± 0.3% |
| | Actinobacteria | .91 ± .01 | 19 ± 0.6% | Actinobacteria | .45 ± .02 | 22 ± 0.2% |
| | Bacteroidetes | .82 ± .01 | 6.1 ± 0.2% | Bacteroidetes | .35 ± .01 | 7.4 ± 0.1% |
| Node | Actinomycetales | – | 11 ± 0.2% | Actinomycetales | – | 12 ± 0.2% |
| | Lactobacillales | – | 9.1 ± 0.2% | Lachnospiraceae | – | 8.4 ± 0.2% |
| | Gammaproteobacteria | – | 8.0 ± 0.2% | Clostridiales | – | 3.8 ± 0.1% |
| Taxon | Streptococcus | – | 3.1% | Oribacterium | – | 1.0% |
| | Lactococcus | – | 3.1% | Corynebacterium | – | 1.0% |
| | Pasteurella | – | 2.6% | Lactococcus | – | 1.0% |
| | Lactobacillus | – | 2.3% | Streptococcus | – | 1.0% |

*Table 16. Tree-level inference*: `cMD3`. Importance (mean ± std) aggregated by depth (cumulative) and taxon across westernization, age, healthy/disease, body site, and stool disease tasks.

| Struct. | westernized (2) | | | age category (5) | | | healthy/disease (2) | | | body site (6) | | | stool disease (2) | | |
|---|---|---|---|---|---|---|---|---|---|---|---|---|---|---|---|
| | Comp. | Acc | Imp | Comp. | Acc | Imp | Comp. | Acc | Imp | Comp. | Acc | Imp | Comp. | Acc | Imp |
| Depth | ≤0 | .93 | 0.1% | ≤0 | .71 | 1.3% | ≤0 | .61 | 0.3% | ≤0 | .91 | 0.4% | ≤0 | .58 | 0.3% |
| | ≤1 | .94 | 0.9% | ≤1 | .76 | 4.4% | ≤1 | .64 | 1.8% | ≤1 | .98 | 3.2% | ≤1 | .62 | 1.7% |
| | ≤2 | .94 | 1.9% | ≤2 | .77 | 6.4% | ≤2 | .65 | 3.2% | ≤2 | .98 | 5.2% | ≤2 | .62 | 3.2% |
| | ≤3 | .96 | 5.3% | ≤3 | .79 | 11% | ≤3 | .67 | 6.4% | ≤3 | .98 | 10% | ≤3 | .65 | 6.4% |
| | ≤4 | .96 | 9.1% | ≤4 | .79 | 18% | ≤4 | .68 | 12% | ≤4 | .99 | 14% | ≤4 | .66 | 12% |
| | ≤5 | .96 | 24% | ≤5 | .80 | 35% | ≤5 | .68 | 31% | ≤5 | .99 | 23% | ≤5 | .66 | 31% |
| | ≤6 | .97 | 100% | ≤6 | .80 | 100% | ≤6 | .68 | 100% | ≤6 | .99 | 100% | ≤6 | .67 | 100% |
| Taxon | Prevotella | – | 1.7% | Blastocystis | – | 0.6% | Lachnoclost. | – | 0.6% | Bacillus | – | 1.4% | Lachnoclost. | – | 0.6% |
| | Murimonas | – | 1.4% | Agathobaculum | – | 0.5% | Bifidobact. | – | 0.3% | Cutibacterium | – | 1.2% | Gemella | – | 0.4% |
| | Tissierellia | – | 1.4% | Staphylococcus | – | 0.5% | Enterococcus | – | 0.3% | Malassezia | – | 1.1% | Bifidobact. | – | 0.3% |
| | Bacteroides | – | 1.4% | Ruminococcus | – | 0.5% | Coprococcus | – | 0.3% | Arcobacter | – | 1.1% | Coprococcus | – | 0.3% |

*Table 17. Tree-level inference*: `DISCO`. Importance (mean ± std) aggregated by depth (cumulative), subtree, node, and cell type across COVID-19, leukemia, and HCC tasks.

| Structure | COVID-19 (blood) | | | leukemia (blood) | | | HCC (liver) | | |
|---|---|---|---|---|---|---|---|---|---|
| | Component | Acc | Imp | Component | Acc | Imp | Component | Acc | Imp |
| Depth | ≤0 | .50 | 6.9 ± 0.3% | ≤0 | .62 | 4.6 ± 0.4% | ≤0 | .88 | 8.1 ± 0.7% |
| | ≤1 | .56 | 11 ± 0.4% | ≤1 | .88 | 26 ± 0.5% | ≤1 | .91 | 38 ± 1.5% |
| | ≤2 | .64 | 20 ± 0.6% | ≤2 | .91 | 43 ± 1.4% | ≤2 | .91 | 55 ± 0.8% |
| | ≤3 | .73 | 50 ± 0.5% | ≤3 | .94 | 68 ± 0.9% | ≤3 | .91 | 77 ± 0.5% |
| | ≤4 | .77 | 76 ± 0.8% | ≤4 | .94 | 86 ± 1.1% | ≤4 | .91 | 96 ± 0.3% |
| | ≤5 | .77 | 97 ± 0.2% | ≤5 | .94 | 98 ± 0.2% | ≤5 | .91 | 99 ± 0.1% |
| | ≤6 (all) | .77 | 100% | ≤6 (all) | .94 | 100% | ≤6 (all) | .91 | 100% |
| Subtree | Immune cell | .80 | 93 ± 0.3% | Immune cell | .93 | 95 ± 0.4% | Immune cell | .92 | 79 ± 1.4% |
| | Epithelial cell | .51 | 0.0% | Fibroblast | .50 | 0.0% | Endothelial cell | .80 | 7.8 ± 1.0% |
| | Fibroblast | .50 | 0.0% | Epithelial cell | .51 | 0.0% | Epithelial cell | .78 | 4.1 ± 0.4% |
| Node | T cell | – | 23 ± 0.6% | Immune cell | – | 22 ± 0.5% | Immune cell | – | 19 ± 1.0% |
| | CD4 T cell | – | 18 ± 0.8% | T cell | – | 16 ± 0.4% | Hema. precursor | – | 9.6 ± 0.8% |
| | T/NK cell | – | 10 ± 0.6% | T/NK cell | – | 14 ± 0.3% | Endothelial cell | – | 7.8 ± 1.0% |
| | B cell | – | 7.2 ± 0.6% | Myeloid cell | – | 7.9 ± 0.5% | B cell | – | 7.0 ± 0.4% |
| Cell type | Treg cell | – | 8.5% | Cycling T/NK cell | – | 11.3% | GMP | – | 4.4% |
| | Cycling T/NK cell | – | 6.5% | MAIT cell | – | 5.0% | Intermediate erythroblast | – | 4.2% |
| | Tfh cell | – | 6.2% | Naive CD8 T cell | – | 4.1% | Late erythroblast | – | 4.2% |
| | Plasma cell | – | 4.8% | Gamma delta T cell | – | 3.2% | MHCII high monocyte | – | 3.6% |

