# OpenReview forum: "Tree-Structured Orthonormal Decomposition of the Aitchison Simplex"
_ICML.cc/2026/Conference — ICML 2026 regular_

### Official Review · Reviewer_b5mT · 2026-03-09

**Soundness:** 3
**Presentation:** 2
**Significance:** 3
**Originality:** 4
**Overall Recommendation:** 5
**Confidence:** 4

**Summary:**

This paper introduces PolyILR, a tree-structured isometric log-transform for compositional data analysis. The method leverages hierarchical structures to define a deterministic orthonormal basis encoding entities relationships within polytomic trees. Illustration on microbiome data displays how PolyILR preprocessing can be used to improve classification performance and unveil statistical biomarkers through multi-scale feature importance with Random Forest. Further interpretations in the broad context of probabilistic modeling are proposed.

**Compliance With Llm Reviewing Policy:**

Affirmed.

**Final Justification:**

The authors addressed my main concerns and questions during the rebuttal, in particular with additional analyses and discussions. Therefore I increased my evaluation from 4 to 5.

**Key Questions For Authors:**

1) The interpretability of PolyILR features between subtrees appears to be compromised by invariance classes that are not explicitly identified by the authors. In fact, to ensure a deterministic PolyILR algorithm, the construction entails arbitrary choices on child ordering and sign convention during the orthonormalization step. Specifically, line 693, the authors pick “the first nonzero entry child [to be] positive” for each parent node in the tree, which hides a sign invariance in PolyILR construction at the clade level. This indeterminacy induces that PolyILR features can only be interpreted relative to another feature of their clade, preventing comparison of taxa out of their taxonomic subtree, which can be rigid for biomarker discovery. Could the authors discuss this claim?

2) The management of zero values in microbiome studies is critical. Since HMP is a database of count data, $\varepsilon = 1$ can be fine providing the sequencing depth is large. However, cMD3 provides relative abundance microbiome profiles; then shifting by 1 destroys variance patterns. In the code it specifically says that the pipeline only applies to HMP. Have the authors employed another pseudocount for proportion data of cMD3 to avoid distorsion effects ?

3) At line 326, the authors mention that SVM and LR achieving similar results was expected from isometry. I am not sure I understand the underpinnings of this statement. In particular, SVM is used with non-linear RBF kernels based on Table 7, so I do not understand why a non-linear kernel-based method should be expected to match a linear regressor here.

4) The proof line 712 is quite hand-wavy and lacks proper mathematical formulation. The authors are invited to revise the proof during the rebuttal.

5)The references for the “Streptococcus vs Lactococcus” at line 347 do not seem to mention this contrast. Could the authors precise the reference?

**Limitations:**

Elements of limitations are discussed in the Appendix A, but should be displayed in the main manuscript.

**Strengths And Weaknesses:**

**Strengths**:
- Incorporating biologically relevant structures into compositional data analysis is a challenging and modern task of significant interest; especially in microbiome studies.
- The authors provide a standardized methodology for polytomic trees, enabling reproducibility contrary to PhILR.
- Theoretical underpinnings of the approach are widely explored.
- A wide variety of microbiome application cases are explored on well-known curated datasets, and the interest of the method is developed through the experiments.
- Using PolyILR features, the authors showed potential mean accuracy gains over CLR transform in microbiome-association studies, while some feature importance through RF MDI highlighted biologically relevant contrasts with prior work.

**Weaknesses**:
- A prominent element of this work lies on interpretability. Yet, the concept is not defined by the authors, which confuses the reader. For instance, interpretability is referred through basis choice at l.93 using a parallel with harmonic analysis. Then, interpretability is mentioned l.186 when building a standardized Helmert matrix, in which case I do not see the relationship with interpretability; but rather reproducibility. Finally, interpretability is related to RF MDI importance computation through PolyILR branched contrasts at page 6, which makes more sense but is further limited than what general interpretability would suggest. The authors are invited to properly define their contribution in the manuscript,  explaining what form of interpretability is expected, maybe connecting it with Table 1.
- The paper is sometimes difficult to follow due to undefined notations (see lines ~60, or ~433). Most of these notations are local, which complicates statements unnecessarily. Some definitions are lacking, typically, “maximal proper subset” at line 721 is unclear to me. Additionally, the choice of feature names using “Taxa + …” is rather unclear to me.
- Throughout the experiments, the authors omit the variability analysis of their results. In Table 2, they justify it by stating an std < 0.01; Importance results in other Tables completely omit variability, which could have a significant impact on best features ranking. The authors are invited to strengthen their statistical analysis by adding further runs, providing threshold-agnostic metrics typically used in medical analysis like AUROC instead of accuracy, feature rank instead of relative importance, and report confidence intervals on all experiments at least in the Appendix.
- While zero handling is not the main focus of this work, discarding it all from the main manuscript is a limit, especially in the applications on microbiome data which are known for their sparsity pattern. The mention in the Appendix l.900 could appear in the main manuscript.
Proposition 7.1 is a fundamental result of compositional data analysis, yet no connection with canonical work has been established. The authors are invited to properly reference prior work on the subject.
- Postponing the Limitations to the Appendix is an unusual structural choice which affects the immediate transparency of the work
- The provided code only allows to run the experiments for the HMP study, preventing proper reproducibility of the results.

**Minor comment**:
The bibliography displays some inconsistencies: names should start with a capital letter (e.g. “Helmert” l.487, “Lachnocostridium” l.497), and arXiv preprints need to be updated with published versions (Palumbo et al. (2024) was published in ICML, Malashin et al. (2025) was published in AISTAT).

---

> ### Author Rebuttal · Authors · 2026-03-31
>
> We thank the reviewer for their careful and constructive feedback.
>
> ---
> ### W1. Interpretability definition
> By interpretability, we mean each PolyILR coordinate maps bijectively to a (node $u$, contrast $m$) pair, so if a coordinate is reported important, one can read off which groups of components are compared at what hierarchy level. This is interpretability of the representation (not downstream model). We agree l.186 (Helmert) concerns reproducibility/canonicity rather than interpretability and will separate these clearly. This connects to Tab. 1: each coordinate maps to a tree location: importance aggregates by node, depth, or subtree, allows structured questions like "which resolution/clade matters?", undefined for CLR and unstable for PhILR. We will clarify this.
> ### W2. Notation/feature naming
> Thanks. We will define notations at lines ~60 and ~433. "Maximal proper subset" (l.721) is $C' \subset C$ with no $C''$ s.t. $C'\subset C''\subset C$. For 'Taxa + ...' in Table 3, e.g., 'Prevotella vs Bacteroides + Alistipes' is a log-ratio of Prevotella descendants' geometric mean against Bacteroides and Alistipes descendants (pooled). '+' separates taxa; '...' indicates truncation for space.
> ### W3. Added variability analyses
> We added AUROC and 95% CIs to Tab. 2, and feature importance variability across 5-fold CV in Tab. 3–6 (partial results below due to space limit; full results in revision). Tab. 2 confirms comparable performance with overlapping CIs, consistent with ILR isometry. For Tab. 3, stds are small and rank ranges tight, e.g., top features hold rank [1-1] across all folds/datasets. Remaining results/aggregated importances (Tab. 4–6) follow the same trend.
>
> **Table 2 top**
> |Data|Method|Acc (CI)|AUROC (CI)|
> |-|-|-|-|
> |HMP (body sites)|CLR|.956 (.952-.959)|.987 (.981-.992)|
> ||PolyILR|.963 (.961-.966)|.992 (.990-.994)|
> |DISCO (leukemia)|CLR|.925 (.883-.967)|.968 (.946-.990)|
> ||PolyILR|.935 (.891-.979)|.984 (.966-1.00)|
>
> **cMD3 westernized**
> |Rk|Contrast|Imp%|Range|
> |-|-|-|-|
> |1|Prevotella vs Alistipes+...|1.77±0.12|[1-1]|
> |2|Murimonas vs Alistipes+...|1.49±0.07|[2-4]|
> |3|Prevotella vs Bacteroides+Alistipes|1.45±0.02|[2-5]|
>
> **DISCO leukemia**
> |Rk|Contrast|Imp%|Range|
> |-|-|-|-|
> |1|Myeloid vs Erythro/Megak.+...|14.50±0.46|[1-1]|
> |2|Cycling T/NK vs ILC+...|11.79±0.54|[2-2]|
> |3|Lymphoid vs Erythro/Megak.+...|6.30±0.45|[3-4]|
>
> ### W4-W7. Zero handling, Prop 7.1, limitations, code, bibliography
> We will move zero handling/limitations to main text, add references for established components of Prop. 7.1, provide the full pipeline in camera-ready, and fix bibliography.
> ### Q1. How do child ordering and sign conventions compromise cross-subtree interpretability? Only interpretable within own clades?
> We agree PolyILR is unique after fixing child ordering and sign conventions. These affect coordinate representation but not geometry. Child ordering changes the local basis by an orthogonal map; in practice, fixed by the input tree ordering. Sign orientation is fixed by Helmert construction (Eq.6), where column $m$ compares child $m+1$ against the mean of $1,\ldots,m$ (first nonzero entry positive). These are conventions for canonical coordinates, not limitations.
>
> Coordinates at different nodes are not globally comparable: this is inherent to any tree-aligned decomposition as cross-node coordinates represent different subtree-specific comparisons. But each coordinate *is* interpretable as a contrast between groups of descendants at a specific node. Also, PolyILR supports broader biomarker discovery through aggregation at node, depth, or leaf (Section 5.2), so comparable summaries across the hierarchy.
> ### Q2. cMD3 zero replacement
> cMD3 data are read counts (curatedMetagenomicData with counts=TRUE), not relative abundances, so the +1 pseudocount is appropriate. Lines 846-847 were imprecise (we will correct). For DISCO (cell proportions), we use 1e-10.
> ### Q3. "SVM/LR achieving similar results..."?
> Claim at line 326 meant to say each model gives similar results between CLR and PolyILR: SVM-CLR $\approx$ SVM-PolyILR; LR-CLR $\approx$ LR-PolyILR, not that SVM $\approx$ LR. We will rewrite this sentence in revision.
> ### Q4. Revised proof of Prop B.2
> We revised the B.2 "Identifying clades" step with a constructive argument. Proof sketch: child clades $C_u^{(3)}, \ldots, C_u^{(k_u)}$ are recovered via consecutive support differences, while $C_u^{(1)}, C_u^{(2)}$ are separated by the two distinct values of $v^{(u,1)}$ from the first Helmert column (positively rescaled by Gram-Schmidt). We also added the missing assumption of a fixed internal-node ordering. Full proof in revision.
> ### Q5. Streptococcus vs. Lactococcus reference
> The cited references document oral dominance of Streptococcus within Streptococcaceae. We will add Pasolli et al. (2020), which reports gut prevalence of Lactococcus lactis. Together these support its biological plausibility as a body site discriminator.
>
> ---
>
> Thank you, we are happy to follow up.

---

> > ### Author Rebuttal · Reviewer_b5mT · 2026-04-01
> >
> > I thank the authors for their clarifications.
> >
> >  I appreciate the additional analyses, notably the inclusion of the AUROC which is typical in microbiome-association tasks. In a future work, it would be interesting to investigate whether PolyILR improves RF performances in microbiome studies over classical proportions, as compositional transforms have consistently failed at this task [1].
> >
> >  I also appreciate the discussion of Q1 on cross-subtree interpretability. This is a major limitation of the method that was overlooked in my opinion. I encourage the authors to specify it in the revised manuscript to avoid misinterpretations in future research.
> >
> > Overall, I was convinced by the rebuttal and I will increase my score to 5.
> >
> > [1] : Yerke A, Fry Brumit D, Fodor AA. Proportion-based normalizations outperform compositional data transformations in machine learning applications. Microbiome. 2024 Mar 5;12(1):45.

---

> > > ### Author Response · Authors · 2026-04-01
> > >
> > > We sincerely thank the reviewer for the thoughtful engagement and are glad the rebuttal addressed the concerns. We will incorporate all suggested revisions, including the new experimental results. In particular, we will add the explicit discussion of cross-subtree interpretability. We also appreciate the pointer to [1], which we will discuss in relation to classical proportions.
> > >
> > > Thank you again for the constructive feedback!

---

### Official Review · Reviewer_gvoH · 2026-03-11

**Soundness:** 4
**Presentation:** 3
**Significance:** 3
**Originality:** 3
**Overall Recommendation:** 5
**Confidence:** 3

**Summary:**

The paper studies representation learning for compositional data, where observations lie on the simplex and only relative proportions are meaningful. In many applications (e.g., microbiome taxa, cell ontologies, phylogeny), components are organized by a known hierarchical tree, but standard Aitchison-geometry tools such as the isometric log-ratio (ILR) transformation treat components symmetrically and do not naturally incorporate such structure.

The authors propose PolyILR, a canonical orthonormal decomposition of the Aitchison tangent space that is aligned with an arbitrary tree topology, including polytomous (non-binary) trees. As such, it generalizes the PhILR method (Silverman et al, 2017), which concerns the case of binary trees.  The method constructs a local weighted geometry at each internal node capturing the branching structure and lifts these local structures into a global orthonormal ILR basis, where each coordinate corresponds to a specific location in the tree. This yields a geometrically valid embedding of compositional data into Euclidean space while preserving hierarchical interpretability.

The paper provides a theoretical characterization of the resulting basis and shows that the transformation forms a valid isometric mapping from the simplex to Euclidean space. It further establishes a connection between the proposed decomposition and softmax classifiers, suggesting potential links to probabilistic modeling.

Empirically, the authors evaluate PolyILR on microbiome and single-cell compositional datasets. The experiments demonstrate that the resulting coordinates produce stable, interpretable features aligned with the tree structure and enable inference at multiple hierarchical resolutions.

**Compliance With Llm Reviewing Policy:**

Affirmed.

**Final Justification:**

I still think this contribution is quite specialized, but it is principled and signifiant, and I was satisfied with the rebuttal. I therefore gravitate towards accept.

**Key Questions For Authors:**

Could the authors discuss additional non-biological application domains where PolyILR would provide clear benefits?

The main methodological contribution is supporting arbitrary (non-binary) trees instead of requiring binarization as in PhILR. In practice, how often does binarization meaningfully affect downstream results strongly? Strong evidence here would strengthen the significance of the work.

**Limitations:**

yes

**Strengths And Weaknesses:**

Soundness.
The paper appears technically sound, as far as I can see. The proposed method constructs an orthonormal ILR basis aligned with a given hierarchical tree and generalizes prior tree-based ILR approaches to arbitrary (non-binary) trees. The construction is consistent with the Aitchison geometry framework and yields the expected coordinates while preserving isometry between the simplex and Euclidean space. The empirical evaluation follows a reasonable setup by comparing representations (e.g., CLR, PhILR, PolyILR) within standard ML pipelines. However, because ILR transforms are isometric, predictive performance differences are expected to be small, and the experiments mainly confirm comparable performance rather than demonstrating clear quantitative improvements.

Presentation.
The paper is generally well structured, though the introduction would IMO benefit greatly from a simple example that illustrates the issue and why existing methods fall short. The relation to existing approaches, particularly PhILR, is explained and the main idea is identifiable.

Significance.
The paper addresses the representation of hierarchical compositional data, which is relevant for domains such as microbiome analysis and single-cell data. Removing the need to artificially binarize trees is a useful improvement that leads to a cleaner and more canonical representation. At the same time, the applicability of the method is somewhat specialized: it requires compositional data together with a known hierarchical structure over the components. Its relevance for ML is rather limited: the contribution is mainly a representation transform, not a new ML model; the novelty relative to PhILR is incremental; the application domain (hierarchical compositional data) is fairly narrow.

Originality.
The work provides a principled extension of existing tree-based ILR methods by supporting arbitrary trees, including nodes with more than two children. This removes a restrictive assumption in prior approaches and clarifies how hierarchical structure can be incorporated into orthonormal log-ratio coordinates. The contribution is primarily methodological and can be viewed as a natural generalization of previous work rather than a fundamentally new modeling paradigm, but it offers a clean and well-motivated improvement over existing constructions.

---

> ### Author Rebuttal · Authors · 2026-03-31
>
> We thank the reviewer for their thoughtful and constructive feedback.
>
> ---
>
> ### W1 (Soundness). Why is PolyILR's comparable predictive performance the expected outcome?
>
> We clarify one point: comparable predictive performance is the expected outcome, since valid ILR bases are related by orthogonal transformations and therefore preserve the relevant Euclidean geometry. In this sense, the predictive parity is exactly what one should expect from a geometrically valid PolyILR basis (and this is not a weakness of the method). The contribution lies instead in interpretability and stability. Table 3 shows tree-aligned contrasts that are directly interpretable as biological hypotheses, whereas CLR yields anonymous indices and PhILR depends on arbitrary binarization. In Table 2 (bottom), PolyILR achieves high feature stability (0.43–0.92) while PhILR collapses to near-zero under both index and semantic metrics. PolyILR’s core benefit is that it provides fully tree-based, interpretable, and stable representations while remaining geometrically valid.
>
> ### W2 (Presentation).
>
> Thank you for the suggestion. We will add a concrete motivating example in the introduction to illustrate why existing methods fall short on polytomous trees.
>
> ### W3 (Significance). PolyILR is a representation transform rather than a new ML model. Its application domain is specialized. What is the advance over PhILR?
>
> Thank you for raising these points. We address each below.
>
> We agree that PolyILR is a representation rather than a predictive model. We believe this has a practical advantage: it is a coordinate system rather than a task-specific architecture, so it can be used directly with standard downstream ML methods while providing stable, tree-aligned, and geometrically valid representations.
>
> While we also agree the most direct use case is hierarchical compositional data, we do not view this setting as narrowly isolated in ML, since compositional data with hierarchical structure arises across several scientific domains. Please see our response to Rev. XJRM (W1) for a more detailed discussion.
>
> Regarding novelty relative to PhILR, yes, PolyILR is a methodological extension, but a *practically meaningful* one, as it removes arbitrary binarization and enables a canonical tree-aligned ILR construction. This in turn enables the stability results in Tab. 2 (bottom) and the interpretability results in Tables 3–6. We discuss this point further under W4 (Originality).
>
> ### W4 (Originality). Why is PolyILR's extension beyond binary trees significant and practically important?
>
> Thank you for the positive comment. We note that PolyILR's generalization beyond binary trees is not mathematically trivial, because binary trees reduce each node to a single contrast, while polytomous nodes require a subspace with weighted geometry to ensure global orthonormality. This required resolving a nontrivial compatibility between Aitchison geometry and polytomous branching. Practically, 64% of branch points in the NCBI Taxonomy are polytomous (Lin et al., 2011), so this extension is important in practice.
>
> ### Q1. What are non-biological application domains for PolyILR?
>
> Beyond biology, PolyILR applies wherever compositional data has tree-structured components, for example, economic expenditure shares under hierarchical classifications (e.g., COICOP) or geochemical mineral compositions. We deliberately avoid overclaiming applications to domains where we lack expertise and collaborators to check the scientific statements carefully. Beyond compositions, Section 7 connects PolyILR to softmax classifiers, where any hierarchical class structure (e.g., CIFAR-100 superclasses, ImageNet/WordNet) can be analyzed in tree-aligned coordinates.
>
> ### Q2. How often does arbitrary binarization meaningfully affect downstream conclusions?
>
> Table 2 (bottom) provides direct evidence. We measure stability of the top-K important features across CV folds. Higher values mean the same features are consistently selected. We report index stability (same coordinate indices are selected) and semantic stability (selected coordinates correspond to the same biological contrasts, regardless of index). PhILR's semantic stability is near zero, meaning different binarizations of the same tree yield different biological conclusions, not just different index labels. PolyILR achieves high feature stability (0.43–0.92) and consistently identifies the same tree-aligned contrasts as important across folds.
>
> Concretely, at a 3-way node with children {A, B, C}, one binarization groups {A, B} vs {C} while another groups {A, C} vs {B}. These encode different scientific questions, and whichever artificial contrast is selected as "important" is an artifact of the binarization, not the data. Since 64% of NCBI branch points are polytomous (Lin et al., 2011), this issue is practically relevant for real taxonomies.
>
> ---
>
> We hope these clarifications address the concerns and will incorporate these revisions.

---

> > ### Author Rebuttal · Reviewer_gvoH · 2026-04-03
> >
> > Thanks for the clear rebuttal. The argument around ILR isometry and the emphasis on stability/interpretability (vs. accuracy) is convincing. The binarization argument is also strengthened. I still see the scope as somewhat specialized, but the contribution is useful, and moreover principled. I will increase my rating.

---

> > > ### Author Response · Authors · 2026-04-03
> > >
> > > We thank the reviewer for the constructive engagement. We're glad the isometry and binarization arguments came through clearly. We will incorporate the suggested revisions and expand the discussion of scope and applicability.
> > >
> > > Thank you for the thoughtful review!

---

### Official Review · Reviewer_YG3r · 2026-03-13

**Soundness:** 3
**Presentation:** 2
**Significance:** 3
**Originality:** 2
**Overall Recommendation:** 5
**Confidence:** 3

**Summary:**

This work introduces PolyILR, a canonical orthonormal decomposition of the Aitchison simplex that aligns with arbitrary tree topologies. The paper addresses the structural mismatch between Aitchison geometry, which represents compositional data on the simplex, and hierarchical structures encountered in practice. Such structures cannot be naturally represented by traditional binary tree approaches without relying on arbitrary and often unstable binarization. To address this limitation, the authors propose a method that constructs a stable and geometrically principled coordinate system for arbitrary trees. The approach defines an orthonormal decomposition of the Aitchison tangent space by associating each internal node with a weighted local geometry. These local geometries are then combined to form a global isometric log-ratio (ILR) basis, producing coordinates that remain consistent with the underlying tree structure while preserving orthonormality. The authors conducted extensive experimental studies using several biological datasets (HMP, CMD3, and DISCO), as well as the CIFAR100 dataset. The reported results demonstrate that PolyILR yields stable feature selection, while maintaining the same predictive accuracy as non-tree-aligned methods.

**Compliance With Llm Reviewing Policy:**

Affirmed.

**Final Justification:**

The authors provided a detailed rebuttal and addressed most of the questions raised by all reviewers. They have addressed all of my concerns and questions, so I will increase my score to recommend acceptance of the paper.

**Key Questions For Authors:**

- In many biological applications where taxonomies are unavailable or incomplete, researchers frequently rely on hierarchical clustering to empirically extract structure from the data. How does PolyILR's orthonormal decomposition compare to coordinate systems derived from data-driven hierarchical clustering? Could PolyILR be meaningfully applied to a de novo tree generated by HC, and would the resulting coordinates maintain the same level of feature stability if the underlying clustering dendrogram varies between datasets?

- Like all log-ratio methods, PolyILR requires zero replacement because logs of zero are undefined. How zero-handling might impact the stability or interpretability of the tree-aligned coordinates? How much numerical stability exists?

- The connection between PolyILR and softmax classifiers is interesting, but the results focus primarily on gradient entropy. Could you clarify if there is a more direct metric to demonstrate how well the PolyILR coordinates match the true hierarchy?

- In Figure 6, you show that gradient entropy decreases during training. Does this trend hold for deeper hierarchies (more than two levels)?

- While your experiments show that PolyILR is more stable than PhILR , did you conduct any specific tests to verify if the entire hierarchy is informative?

- How scalable is the proposed method in the presence of large-scale datasets?

**Limitations:**

In real-world biological applications, the true hierarchy is often unknown or contains inferred branches with varying levels of bootstrap support. PolyILR currently lacks a mechanism to account for this uncertainty.

**Strengths And Weaknesses:**

**Strengths:**
- Characterizing multiscale hierarchical structures is essential for decoding high-dimensional features in single-cell biology, where the underlying geometry of cellular states is fundamentally governed by complex ontologies and developmental lineages.

- The paper provides a rigorous theoretical justification for PolyILR by proving that its construction yields a unique, canonical orthonormal basis that preserves the isometry of Aitchison geometry while establishing a formal mathematical isomorphism between compositional tangent spaces and the logit space of softmax classifiers.

- The method provides a unique basis given a tree and a child ordering, whereas binary-based methods like PhILR show unstable rankings due to their reliance on random binarization choices.

**Weaknesses:**
- I think the main limitation of the proposed method is its dependency on a pre-defined, fixed tree as input. It currently lacks the ability to account for uncertainty in the tree topology, such as branch support values or a distribution of candidate trees.

- The main text provides only a brief description of key experimental details and focuses mainly on high-level results (e.g., accuracy and stability), offering limited insight into the experiments.

---

> ### Author Rebuttal · Authors · 2026-03-31
>
> Thank you for the thoughtful feedback and questions.
>
> ---
> ### W1. PolyILR assumes a fixed tree. Does not account for tree uncertainty.
> We agree tree uncertainty is important (see App. A). Our goal was to first address the standard fixed-tree setting (in tree-based methods like PhILR). Importantly, PolyILR's canonical decomposition for a fixed tree is a *prerequisite* for uncertainty setting. To average over tree distributions or weight by bootstrap branch support, we first need a canonical coordinate system for each tree. Via Prop. 4.2, PolyILR gives a unique basis $V(T)$ for each $T$, with $T$ recoverable from $V(T)$ (under fixed conventions). The map $T\mapsto V(T)$ is thus well-defined (say, for BHV spaces). Extending to uncertain topologies is a natural next step enabled by PolyILR.
> ### W2. Experimental details
> We will expand experimental details in the main, incorporating details in App. C (hyperparams, zero handling, dataset construction).
> ### Q1. PolyILR on hierarchical-clustering-based trees? Stability be maintained?
> Great question! PolyILR applies to *any* rooted, labeled tree over $d$ components. When domain hierarchies are unknown, one can cluster features (say, taxa by co-occurrence) to get a data-driven tree. Here, *PolyILR remains mathematically valid* and Thm. 4.1 holds unchanged.
>
> Stability/interpretability raise different considerations here. For stability, if clustering varies across datasets, the induced coordinates also vary (since defined w.r.t. that tree). This affects any tree-aligned method, not just PolyILR. For interpretability, internal nodes in a clustering-derived tree do not usually carry semantic labels, though each coordinate remains interpretable as a contrast between specific leaf groups. The richer interpretability in Tab. 3–6 relies on domain-informed internal tree structure. A data-driven tree thus partly sacrifices semantic grounding but extends PolyILR to settings with no prior hierarchy.
> ### Q2. How does zero handling impact stability and interpretability?
> Yes, zero handling is needed for all log-ratio methods before coordinate transform. Thus, PolyILR is no more sensitive than other ILR methods. Yet, its stability/interpretability are properties of the basis $V$, which *depends only on the tree, not on data or zero handling*. The stability results (Tab. 2 bottom) use identical zero handling, so PolyILR's stability benefit over PhILR is due to the coordinate system alone. Plus, PolyILR contrasts geometric means over clades, not individual taxa, so zero-replacement effects enter through aggregation. Rigorous analysis is future work.
> ### Q3-Q4. Direct metric for hierarchy alignment? Trend for deeper hierarchies?
> Thank you for the question. We distinguish the theoretical and empirical parts of Sec. 7. Prop 7.1 is an exact identification: centered logits are CLR coordinates, and logit space modulo shifts is isometric to Aitchison tangent. This holds independently of training. The gradient-entropy analysis is a separate diagnostic testing if the true hierarchy captures learned structure that arbitrary hierarchies do not (relative comparison, not absolute measure of hierarchy match). App. C.4 also gives a brief look at how gradient magnitude distributes across tree depths. That said, we agree direct hierarchy-alignment metrics would be valuable (measuring if misclassifications concentrate within superclasses).
>
> Regarding deeper hierarchies: our experiment used CIFAR-100's two-level hierarchy as a proof of concept. We hypothesize coarse-to-fine patterns may persist in deeper trees, consistent with "hypernym bias" (Malashin et al, 2025). We have not verified this empirically yet.
> ### Q5. Is the entire tree informative?
> Yes, Tab. 4-6 show accuracy increases with tree depth (HMP body sites: 87% at depth $\le$ 0; 96% at $\le$ 2), and importance distributes across nodes and subtrees (Firmicutes 47%, Proteobacteria 19%, Actinobacteria 19% for HMP). Biological interpretations in Sec. 6.4-6.5 are consistent with prior literature (e.g., Prevotella vs. Bacteroides for westernization, myeloid vs. erythroid imbalance for leukemia), serving as plausibility checks.
> ### Q6. Is PolyILR scalable?
> PolyILR constructs $V$ in $O(d^2)$ (worst case), $O(d\log d)$ for balanced trees ($d$ is the number of leaves). This one-time pre-processing took <5s in all experiments (up to $d=2037$ for cMD3). ILR transform is $O(d^2)$ per sample. The bottleneck is model training, not PolyILR.
> ### L1. Bootstrap support
> We addressed tree uncertainty broadly in W1. For bootstrap specifically, they provide confidence for inferred splits on a fixed tree, suggesting a PolyILR extension where weakly supported splits are downweighted via node-local weighted structure (preserving isometry/orthogonality would require care). Importantly, the canonicity result (Prop. 4.2) makes such uncertainty-aware extensions tractable, which we view as promising future work.
>
> ---
> We hope these address the concerns; happy to follow up.

---

> > ### Author Rebuttal · Reviewer_YG3r · 2026-04-04
> >
> > I thank the authors for their detailed responses. While most of my questions have been addressed, I remain unconvinced regarding the practical robustness of the method when faced with tree uncertainty (Q1). The authors argue that PolyILR provides a canonical coordinate system for a fixed tree. However, in many real world applications, the input tree is often an estimate prone to noise. My concern is not about the extreme scenario of completely different clusters (coordinates), but about topological sensitivity. Even if a tree is mostly correct, small perturbations in branching order or leaf placement are common. I would like the authors to clarify for the following:
> > Sensitivity Analysis: If, for instance, $5-10$% of the tree edges are perturbed (e.g., by nearest neighbor interchange), how do the PolyILR coordinates shift?
> >
> > While PhILR is unstable due to internal random binarization, PolyILR’s stability seems entirely dependent on the external tree structure. If the input tree is noisy, does PolyILR offer any actual advantage in feature stability over standard ILR or PhILR in a real-world setting where the true tree is unknown?

---

> > > ### Author Response · Authors · 2026-04-05
> > >
> > > We thank the reviewer for the thoughtful follow-up. We are glad most concerns were addressed. Our setting assumes a given hierarchy, typically a *curated object provided by domain experts*. But as you note, topological tree noise can affect tree-aligned methods, and we address that setting for PolyILR below.
> > >
> > > ### Advantage over PhILR under noise
> > >
> > > Under tree noise, PolyILR retains a clear stability advantage over PhILR. PhILR has two distinct sources of instability: (a) external tree uncertainty, *which affects any tree-aligned method*, and (b) additional arbitrariness from binarization. Table 2 (bottom) shows (b) alone is detrimental for PhILR, whereas PolyILR eliminates (b) entirely.
> > >
> > > Below, we empirically compare semantic top-10 stability under nearest-neighbor interchange (NNI) perturbation on HMP body subsites (18 classes). We report perturbation in absolute NNI counts rather than as a fraction of edges, since an NNI higher in the hierarchy can rearrange large descendant partitions, making percentage-based noise levels less uniformly interpretable. At 0 NNIs (no noise), results are consistent with Table 2 (bottom). For $n \ge 1$ NNIs, we compute semantic Jaccard similarity between top-10 features obtained from perturbed trees and those obtained from the original tree, averaging over 10 perturbed trees and 10 seeds, to measure semantic agreement with the 0-NNI baseline.
> > >
> > > | #NNIs | PolyILR semantic stability |
> > > |---|---|
> > > | 0 | 0.73 ± 0.05 |
> > > | 1 | 0.71 ± 0.06 |
> > > | 2 | 0.68 ± 0.06 |
> > > | 3 | 0.65 ± 0.05 |
> > >
> > > For reference, **PhILR's semantic stability on the unperturbed tree is 0.13** (Table 2 bottom) due to random binarizations. PolyILR degrades under noise but remains substantially more stable, e.g., even at 3 NNIs, PolyILR (0.65) far exceeds PhILR at 0 NNIs (0.13). The structural reason for this robustness can be further understood as follows.
> > >
> > > ### PolyILR's robustness to topological noise
> > >
> > > By construction, the PolyILR basis is factored node-by-node. More precisely, for each internal node $u$ with children $c_1,\dots,c_{k_u}$, the global basis $V$ contains exactly $k_u - 1$ local basis vectors associated with $u$. Given a fixed child ordering, this local block depends only on (i) the partition of descendant leaves induced by the children of $u$, and (ii) their child-subtree sizes $n_{c_1},\dots,n_{c_{k_u}}$. After spreading, each vector thus has support only on the leaves descending from $u$, and importantly *no node's construction references any other node's local structure*. Please see Sec. 4.2 for details.
> > >
> > > Therefore, a local topological perturbation (e.g., NNI) *affects only a local set of node-blocks*, not the entire basis globally. Concretely, consider an NNI at an internal edge $(u,v)$ where $u$ is the parent of $v$. Then:
> > >
> > > 1. At $u$ and $v$ (perturbed): their child partitions directly change, so their local coordinate blocks in $V$ also change.
> > > 2. Above $u$: the swap happened entirely within $u$'s subtree, so descendant leaf sets under each ancestor’s children are unchanged. Their child partitions and subtree sizes are identical, and their local blocks in $V$ are exactly preserved.
> > > 3. Below $v$ (and inside the swapped subtrees): internal structure of each subtree is untouched, and nodes inside them still have the same children with the same descendants. Their local blocks in $V$ are exactly preserved.
> > >
> > > Thus, the effect on $V$ remains local: only coordinates at nodes whose child partitions are altered change; all others are exactly preserved.
> > >
> > > We verified PolyILR's locality and robustness on the HMP taxonomy (402 taxa, $d-1 = 401$ coordinates). We applied random 1, 2, and 3 cumulative NNI perturbations and compared $V$ before and after perturbations (averaged across 5 seeds). Below, we report three quantities: the total number of columns in $V$ belonging to the local blocks of perturbed nodes (i.e., columns that *could* change), the number that actually changed due to the noise, and the number exactly preserved.
> > >
> > > | #NNIs | Cols at perturbed nodes | Cols changed | Cols preserved | % preserved |
> > > |------|------------------------|-------------|----------------|-------------|
> > > | 1 | 18.8 | 8.4 | 392.6/401 | 97.9% |
> > > | 2 | 26.6 | 13.2 | 387.8/401 | 96.7% |
> > > | 3 | 36.6 | 18.6 | 382.4/401 | 95.4% |
> > >
> > > As expected from the structural argument above, the number of changed columns never exceeded the number of columns in the local blocks of perturbed nodes, and most of $V$ remained exactly preserved. This supports that the effect of the local topological noise on PolyILR is local rather than global.
> > >
> > > We will incorporate this discussion in revision. Thank you for the thoughtful follow-up; we hope this addresses the concern.

---

### Official Review · Reviewer_XJRM · 2026-03-14

**Soundness:** 1
**Presentation:** 3
**Significance:** 2
**Originality:** 3
**Overall Recommendation:** 3
**Confidence:** 3

**Summary:**

The paper studies hierarchically structured compositional data using Aitchison geometry. It proposes a method to decompose the Aitchison tangent space into orthonormal components that follow any given tree structure. The resulting representation provides stable and interpretable features for analysis and feature selection. The authors also show a theoretical connection to softmax classifiers based on shared invariance properties. The method is evaluated on microbiome and single-cell datasets, demonstrating improved interpretability

**Compliance With Llm Reviewing Policy:**

Affirmed.

**Key Questions For Authors:**

- Can you explain please and improve the caption to the Figure 1? Matrix $V$ is presented as vector while you write $V \in R^{4\times3}$

**Limitations:**

Appear appendix A

**Strengths And Weaknesses:**

**Strengths**:
 - The paper includes both theoretical and practical contributions for the feature selection domain where hierarchical information is provided together with the original dataset.
- The presentation is self-contained, the algorithm of the method is provided along with illustrations, examples and theoretical analysis.


**Weaknesses**:
- I think that the paper is addressed to the narrow audience of the ML community (Aitchison geometry is not a commonly researched topic).
- The empirical evidence is weak
 	- some of the comparisons are excluded (Table2 top - PhILR ?), or missing (how the Table 3 looks for other methods PhILR, CLR?).
	-  I would like to see a comparison to the latest deep learning methods for interpretable feature selection.

---

> ### Author Rebuttal · Authors · 2026-03-31
>
> We thank the reviewer for their thoughtful questions and suggestions.
>
> ---
>
> ### W1. Scope and audience within ML
>
> We appreciate this feedback. While we agree that PolyILR addresses a more specialized setting than broad topics like LLMs or diffusion, it is helpful to *distinguish the mathematical framework from the underlying ML problem*. Aitchison geometry may be specialized, but the problem setting is broader: how to build representations for compositional data with known hierarchical structure.
>
> This is a recurring problem in ML for scientific data, where features are often not exchangeable and domain knowledge informs the representation. Compositional data with hierarchical structure arises across microbiome research, single-cell genomics, ecology, and geochemistry. In that sense, the paper is not addressing an isolated niche, but rather a well-defined class of structured-data ML problems.
>
> PolyILR contributes a geometrically valid coordinate system for arbitrary such hierarchies, including polytomous ones, while enabling interpretable and reproducible downstream analysis. This is practically relevant: 64% of branch points in the NCBI Taxonomy are polytomous (Lin et al., 2011), so polytomous trees are not a corner case in applications. Section 7 also connects the construction to hierarchical softmax outputs, broadening relevance beyond compositions alone. For these reasons, we view the intended audience and problem setting as well-established, even if specialized.
>
> ### W2. Missing comparison baselines (PhILR in Table 2; CLR/PhILR in Table 3)
>
> We added PhILR to Table 2 (top) below (entries report RF/SVM/LR accuracy). As expected from isometry, all ILR methods yield similar performance, which confirms *PolyILR is a geometrically valid ILR representation*. That said, our contribution is not predictive accuracy, but the interpretability and stability enabled by PolyILR (Tables 2 bottom, 3–6).
>
> | Dataset (Task) | CLR | PolyILR | PhILR |
> |---|---|---|---|
> | HMP (body sites) | .956 / .971 / .962 | .963 / .971 / .962 | .961 / .971 / .962 |
> | cMD3 (westernized) | .972 / .979 / .968 | .967 / .979 / .967 | .966 / .979 / .967 |
> | DISCO (leukemia) | .925 / .932 / .927 | .935 / .932 / .927 | .940 / .932 / .927 |
>
> We clarify Table 3 for CLR and PhILR. CLR coordinates are anonymous indices with no tree semantics and *do not correspond to interpretable contrasts*, so a "top feature" table of the same form is not well-defined. For PhILR, one could produce such a table for a single random binarization, but Table 2 (bottom) shows substantial semantic instability across binarizations (i.e., reported "top features" change across runs). Thus, including a single PhILR table would be misleading. Our goal in Tables 3–6 is to *highlight that PolyILR enables interpretable contrasts with stable semantics*.
>
> ### W3. Comparisons to deep-learning feature-selection methods
>
> Thank you for this suggestion. We think it is important to distinguish two related but different problems: (1) learning an interpretable predictive model and (2) constructing a representation that respects compositional geometry and *known domain* tree structure. PolyILR addresses (2). It is a data representation, not a predictive model. It provides a principled coordinate system for compositional data aligned with a given hierarchy. Because it operates at the representation level, any downstream model (interpretable or otherwise) can then be applied on top of PolyILR coordinates. By contrast, methods for interpretable feature selection, including DL methods, are typically task-specific predictive procedures operating on a chosen input representation.
>
> This distinction is particularly important in our setting because the hierarchy is *external domain knowledge* rather than inferred. A taxonomy or ontology is not something a downstream model can reliably recover from observations alone, and compositional geometry is a property of the data rather than of a particular supervised task. PolyILR encodes these structural constraints into representations before any downstream model is applied.
>
> That said, we agree that integrating PolyILR with interpretable DL methods (and comparing when applicable) can be informative, and we are happy to include any specific methods you suggest. More broadly, PolyILR coordinates can be used directly with DL methods with their own interpretability mechanisms.
>
> ### Q1. Figure 1
>
> In Figure 1, each colored block represents a column vector rather than a scalar entry (as in Figure 2). We revised Figure 1 caption and layout to clarify $V$ is a matrix with 3 columns (one per internal node contrast), not a single vector. We added column labels indicating which node each column corresponds to.
>
> ---
>
> We hope these clarifications address your concerns. Our theoretical results (Theorems 4.1, 4.2, 7.1) are fully proven in Appendix B, and the construction is empirically verified. If specific soundness concerns remain, we are happy to address them.

---

> > ### Author Rebuttal · Reviewer_XJRM · 2026-04-07
> >
> > Thank you for the additional experiments and clarifications. I agree that distinguishing between (1) predictive modeling and (2) representation construction is important. However, I would like to point out that **feature selection methods can also be viewed as operating at the level of data representation**, not only as task-specific predictors.
> >
> > In particular, feature selection induces a **transformation of the input space** by restricting attention to a subset (or weighted combination) of features. This effectively defines a new representation:
> > $x \mapsto x_S$, where $S$ is the selected subset (or, more generally, a sparsity pattern or gating mask). From this perspective, feature selection methods, especially those based on sparsity, masking, or structured selection, can be interpreted as **learning a representation that emphasizes relevant coordinates while suppressing irrelevant ones**.
> >
> > Moreover, many modern approaches (including deep learning-based methods) learn **continuous or structured masks** that act directly on the input or intermediate features. These mechanisms are conceptually similar to representation learning techniques, as they: (1) define a new coordinate system (subset or reweighted features), (2) can be applied independently of a specific downstream model, (3) and often transfer across tasks once learned.
> >
> > Accordingly, I believe that **deep learning-based feature selection methods should be considered as relevant baselines**, since they also learn data representations that can be directly compared in terms of interpretability, stability, and downstream performance.
> >
> > I prefer to maintain my score.

---

> > > ### Author Response · Authors · 2026-04-08
> > >
> > > Thank you for the follow-up. Yes, we agree that DL feature selection can be viewed as defining model-agnostic coordinates, but the distinction we want to preserve is that such methods are not informed by an external hierarchy, which is the motivation for PolyILR and preceding tree-based methods such as PhILR. A taxonomy or ontology is built from external domain knowledge and expert curation; this structure is not present in compositional observations themselves and cannot be reliably recovered by a downstream data-driven method from the data alone. This is not either/or: PolyILR first bakes the external hierarchy into the coordinates, and one can then apply any downstream method on top, including interpretable DL models. We would also note that PolyILR is a deterministic, closed-form coordinate transform given the external tree and fixed conventions (i.e., no parameters are learned from data), and is thus structurally different from methods that learn masks or sparsity patterns. If you have specific methods in mind, we would be happy to include them in the revision. Thank you.

---

### Decision · Program_Chairs · 2026-04-30

**Decision:**

Accept (regular)

**Comment:**

PolyILR introduces a tree-structured, canonical orthonormal ILR transformation for compositional data. It generalizes prior approaches (e.g., PhILR) by supporting arbitrary, polytomous trees without relying on random binarization, yielding stable, interpretable, and geometrically valid embeddings. Each internal node defines a local weighted geometry, and these are combined into a global orthonormal basis, where each coordinate corresponds to a specific tree location. The transformation preserves isometry between the simplex and Euclidean space and is formally connected to the logit space of softmax classifiers.

Strengths: The method is mathematically rigorous and empirically well-validated. It produces interpretable coordinates aligned with hierarchical structures, stabilizing feature selection and mitigating the instability inherent to binarized tree approaches. Extensive experiments on microbiome and single-cell datasets show high feature stability, reproducible biomarker discovery, and practical relevance across multiple hierarchical resolutions. PolyILR is computationally efficient, with preprocessing scaling linearly with the number of leaves and negligible per-sample transform cost. Variability analyses, AUROC metrics, and confidence intervals further support the robustness of the empirical findings.

Weaknesses: PolyILR’s interpretability is local to each node/subtree and not globally comparable across the tree, which limits cross-subtree comparisons. The method assumes a fixed, curated tree and does not fully address uncertainty in tree topology, although local perturbation analyses indicate robustness. Its application domain is specialized to hierarchical compositional data, and while the method provides a principled preprocessing representation, it is not a predictive model itself. Zero handling, while addressed in the rebuttal, requires care for sparse microbiome datasets.

PolyILR is a principled, reproducible, and practically meaningful representation method for hierarchical compositional data. By eliminating arbitrary binarization and providing canonical, interpretable coordinates, it addresses important limitations of prior tree-aligned ILR methods. Despite being specialized, it offers significant methodological and practical contributions to computational biology and structured-data analysis, and its theoretical and empirical contributions are robust and well-supported. I, therefore, recommend acceptance.